# Subgridding High Resolution Numerical Weather Forecast in the Canadian Selkirk range for local snow modelling in a remote sensing perspective

Paul Billecocq[1, 2], Alexandre Langlois[1, 2], and Benoit Montpetit[3]

[1]Département de Géomatique appliquée, GRIMP, Université de Sherbrooke, Sherbrooke, QC, Canada
[2]Centre d'études nordiques, Québec, QC, Canada
[3]Climate Research Division, Environment and Climate Change Canada, Ottawa, ON, Canada

**Correspondence:** Paul Billecocq (paul.billecocq@usherbrooke.ca)

**Abstract.** Snow Water Equivalent (SWE) is a key variable in climate and hydrology studies. Yet, designing a SWE retrieval algorithm is not trivial, as multiple combinations of snow microstructure representations and SWE can yield the same radar signal. The community is converging towards forward modeling approaches using an educated first guess on the snowpack structure. However, snow highly varies in space and time, especially in mountain environments where the complex topography affects atmospheric and snowpack state variables in numerous ways. Automatic Weather Stations (AWS) are too sparse, and high-resolution Numerical Weather Predictions systems have a maximal resolution of 2.5 km × 2.5 km, which is too coarse to capture snow spatial variability in a complex topography. In this study, we designed a subgridding framework for the Canadian High Resolution Deterministic Prediction System. The native 2.5 km × 2.5 km resolution forecast was subgridded to a 100 m × 100 m resolution and used as the input for snow modeling over two winters in Glacier National Park, British Columbia, Canada. Air temperature, relative humidity, precipitation and wind speed were first parameterized regarding elevation using six Automatic Weather Stations. Alpine3D was then used to spatialize atmospheric parameters and radiation input accounting for terrain reflections and perform the snow simulations. Modeled snowpack state variables relevant for microwave remote sensing were evaluated against profiles generated with Automatic Weather Stations data and compared to raw HRDPS driven profiles. Overall, the subgridding framework improves on average the optical grain size (OGS) bias by 18%, and the modelled SWE by 16% with regards to simulations driven with raw HRDPS forecasts. This work could lead up to a 7 dB improvement in the snowpack SAR backscattering modelling, and hence provides the necessary basis for SWE retrieval algorithms using forward modeling in a Bayesian framework.

## 1 Introduction

Seasonal snow governs several feedback loops that directly affect our planet's climate and plays a major role in its hydrological dynamics. With its high albedo, snow reflects a large proportion of the incoming solar radiation, which in return helps to mitigate global warming (IPCC, 2019). Furthermore, snow insulates the underlying soil, affecting the microbial activity, carbon fluxes, and permafrost freeze/thaw cycles (Natali et al., 2019; Biskaborn et al., 2019). Moreover, seasonal snow melt provides

connected watersheds with freshwater, sustaining natural ecosystems and human infrastructure . Finally, extreme precipitation events and resulting snow melt can cause devastating floods (Pomeroy et al., 2016; Vionnet et al., 2020), so that managing
runoff would highly benefit both society and the economy (Sturm et al., 2017).

Yet, snow mass (or Snow Water Equivalent, SWE) remains poorly characterized, especially in mountainous regions where a significant amount of SWE is stored at the continent scale (Wrzesien et al., 2018). Global SWE products inferred from passive microwave observations are available at a 25 km resolution (Luojus et al., 2021), which is too coarse to capture the SWE spatial variability (Derksen et al., 2021), and mountains are simply omitted or masked out. Moreover, both observations
from passive microwaves and modeling efforts yield negative biases when estimating mountain or deep-snow SWE on the global scale (Vuyovich et al., 2014; Wrzesien et al., 2018; Pulliainen et al., 2020). Hence, the snow remote sensing community is promoting active remote sensing, which provides higher spatial resolution information compared to passive microwaves products (Tsang et al., 2022; Rott et al., 2010; Derksen et al., 2021). The sensitivity of the Synthetic Aperture Radar (SAR) signal to SWE has been proven at the Ku-band (King et al., 2015; Lemmetyinen et al., 2016), and recent studies suggest that
C-band could also be used for snow depth retrieval (Lievens et al., 2019, 2022), a key parameter for SWE retrieval, although it is contrasting with previous research (Dozier and Shi, 2000). However, linking SWE to SAR backscattering is not trivial as it does not depend solely on SWE (which is a function of snow height and density), but also on the snow microstructure. Consequently, several combinations of SWE and snowpack microstructures can yield similar backscattering values, creating a non-unique inversion solution (Tsang et al., 2022). As a result, recent inversion algorithms tend towards a Bayesian framework
where a forward scattering model is used to generate possible backscattering values, and the best fitting one is selected using a weighted cost function (Lemmetyinen et al., 2018; King et al., 2018, 2019; Zhu et al., 2021). So far, these studies only paired airborne radar observations with fields measurements, but coupling a radiative transfer model with a snow physics model still has to be explored in the active microwaves domain.

Advanced thermodynamic multi-layered snow models such as Crocus or SNOWPACK produce SWE and microstructure
parameters estimates (Brun et al., 1992; Vionnet et al., 2012; Lehning et al., 2002). Such models can be driven either by Automatic Weather Stations (AWS) measurements, atmospheric models, or reanalysis products. On the one hand, weather stations provide very accurate measurements of the atmospheric conditions at the local scale. However, they need human maintenance, are subject to outages, local biases, and usually undersample the spatial heterogeneity of the processes at stake, especially in complex terrain. As a result, AWS spatial interpolation in mountainous areas is not always accurate (Lundquist
et al., 2019). On the other hand, the High-Resolution Deterministic Prediction System (HRDPS, Milbrandt et al. (2016)) model is known for its negative bias in precipitation, resulting in a negative bias in snow depth and SWE (Bellaire et al., 2011, 2013; Côté et al., 2017).

Several NWP downscaling schemes have already been proposed. Liston and Elder (2006) introduced the MicroMet model which is now widely used by the community, and is part of several more recent models. In MicroMet, a high-resolution DEM
(30 m to 1 km) is used to generate the overlying atmospheric forcing from a coarser grid or a sparse network of Automatic Weather Station. This allows to produce a physically sound downscaling when compared to naive interpolation methods, but

without the need to run a computationally intensive fully dynamic atmospheric model at the local scale. In Micromet, lapse-rates are used for air temperature, dew point temperature (for relative humidity), and precipitation.The algorithm for wind speed takes terrain slope and curvature into account. Incoming solar radiations are split between direct and diffuse radiation and adjusted with cloud cover and terrain shading. Fiddes and Gruber (2014) developed the TopoSCALE model, which can be seen as an iteration over the MicroMet model. The main difference with MicroMet lies in the precipitation subgridding that takes into account wind redistribution by altering the precipitation field with climatology data after applying the lapse-rate correction from Liston and Elder (2006). In the Canadian Hydrology Model (CHM), Marsh et al. (2020) take this idea one step further, adding snow modelling to the atmospheric model subgridding. In this study, the high-resolution DEM used for subgridding is first transformed into an unstructured triangular mesh (or Triangulated Irregular Network, TIN). This allows to reduce the computational cost of the model, as TINs are known to be more efficient at representing terrain than rasters (Marsh et al., 2018). The input meteorology can be either real AWSs, or an array of "virtual stations" extracted from any atmospheric model and defined by latitude, longitude and elevation. All virtual stations are vertically corrected to a common elevation reference using a dedicated set of specialty modules. The virtual station array is then spatialized over the TIN using either Inverse Distance Weighting, thin plate splines with tension, or the nearest station with no interpolation. This parameterized atmospheric forcing is then used to run snow simulations using either iSNOBAL, SNOWPACK, or Crocus (**?**)refs for snow models). Point-scale validation was performed for SWE using both SNOWPACK and Snobal driven by observed atmospheric data and evaluated against SWE field measurements for one snow season. The atmospheric data spatialization capacity was assessed by performing a leave-one-out analysis with the array of AWS. However, the capacity of the model to subgrid NWP was left unassessed. Vionnet et al. (2021) used the CHM with a novel wind-downscaling strategy to subgrid forecasts from the High Resolution Deterministic Prediction System (HRDPS) and simulate snow conditions at 50 m during one snow season using 2-layer Snobal within CHM as the snow model. The modelled snow depth and SWE spatial variability was evaluated against a spring Airborne Laser scanning snow depth survey. With snow hydrology as a main application, it is natural that the evaluation for CHM in Marsh et al. (2020) and Vionnet et al. (2021) is focused on SWE and snow depth. However, for remote sensing applications, specifically SAR signal inversion, snow microstructure and layering is of particular importance (King et al., 2018; Zhu et al., 2021; Tsang et al., 2022). Moreover, remote sensing products are written in a gridded raster format, the TIN mesh used in CHM, although very efficient, becomes an issue when pairing the model's output with satellite imagery. The Alpine3D model is a spatially distributed 3D model, which allows running the vertical 1D multi-layer snow model SNOWPACK over a gridded DEM, considering the spatial processes affecting atmospheric variables (Bartelt and Lehning, 2002; Lehning et al., 2002, 2006). Weather data is spatialized using the MeteoIO library (Bavay and Egger, 2014). However, MeteoIO is geared towards AWS spatialization, and it does not include an atmospheric model subgridding scheme. This highlights the fact there is currently a need in the community for both the design and the evaluation of an atmospheric model subgridding framework to perform snow modelling in a SAR remote sensing coupling context. The following research should be able to answer the following questions:

1. How do subgridded HRDPS forecasts compare to reference Automatic Weather Stations in the simulation domain ?

2. Do the resulting atmospheric forcings lead to an improvement in snowpack modelling, especially for critical snow parameters in remote sensing applications ?

3. Which degree of spatial variability with regards to snow parameters can be reached by such a subgridding framework ?

To try and answer these questions, we first built a subgridding module to downscale HRDPS grids as a Virtual Weather Station array. Second, we spatialized atmospheric parameters and performed snow simulations on the study area using the Alpine3D model over two consecutive winters (2018–2019 and 2019–2020). Weather parameters subgridding and snowpack state parameters were assessed at three reference weather stations using an array of statistical criteria and a Dynamic Time Warping algorithm (Hagenmuller and Pilloix, 2016; Hagenmuller et al., 2018; Herla et al., 2021). Finally, we assessed the spatial variability capacity of the proposed subgridding framework over the whole simulation domain and within one HRDPS grid cell.

## 2 Study area

This study was conducted in the Rogers Pass area of Glacier National Park (GNP), British Columbia, Canada (Figure 1), which is part of the Selkirk range in the Columbia Mountains. The pass is used as a transportation corridor by the Trans Canada Highway and the Pacific Railway, making it the busiest transport corridor in Western Canada (Bellaire et al., 2016). The pass is exposed to 144 avalanche paths, and as a result, Rogers Pass hosts the largest avalanche control operation in Canada (Delparte et al., 2008). The operation has been ongoing since 1965 and the site has been used as a snow research site ever since, making this area the longest record of mountain snow in Western Canada (Fitzharris, 1987; Bellaire et al., 2016; Madore et al., 2022). The study area is 18 km by 16 km wide, covering 288 km$^2$ of complex topography, with elevations ranging from 840 m a.s.l at the valley bottom to 3284 m a.s.l. In winter, the Columbia Mountains snowpack is characterized as a transitional snowpack with a maritime influence. Hence, westerly fluxes coming from the Pacific mainly govern the precipitation pattern in this mountain range. Occasionally, dryer and colder systems from the northeast can also hit the range, bringing some continental influence to the east of the study area. On average, the snowpack reaches 3.2 m at its peak, usually around the end of March and early April.

The Park has seven Automatic Weather Stations (AWS) at different elevations around the Highway corridor. The measured variables are air temperature (TA, °C), relative humidity (RH, %), wind speed (VW, m.s$^{-1}$), wind direction (DW, degrees), precipitation (PSUM, mm), incoming long wave radiation (ILWR, W.m$^{-2}$) and incoming shortwave radiation (ISWR, W.m$^{-2}$). Some stations include a snow height (HS, cm) sensor. Table 1 summarizes the set of meteorological variables available for each AWS. In this study, we used two winter time series: from September to April, 2018–2019 and 2019–2020. The 2018–2019 season had overall colder temperatures and was relatively dry. The 2019–2020 season had milder temperatures and abundant precipitations. As a result, in 2019–2020, the snowpack was deeper and mostly composed of rounded grains where, in 2018–2019, the shallower snowpack and colder temperatures led to a mostly faceted snowpack. Both seasons had rain-on-snow episodes in the early season, which created melt-freeze crusts at the bottom of the snowpack.

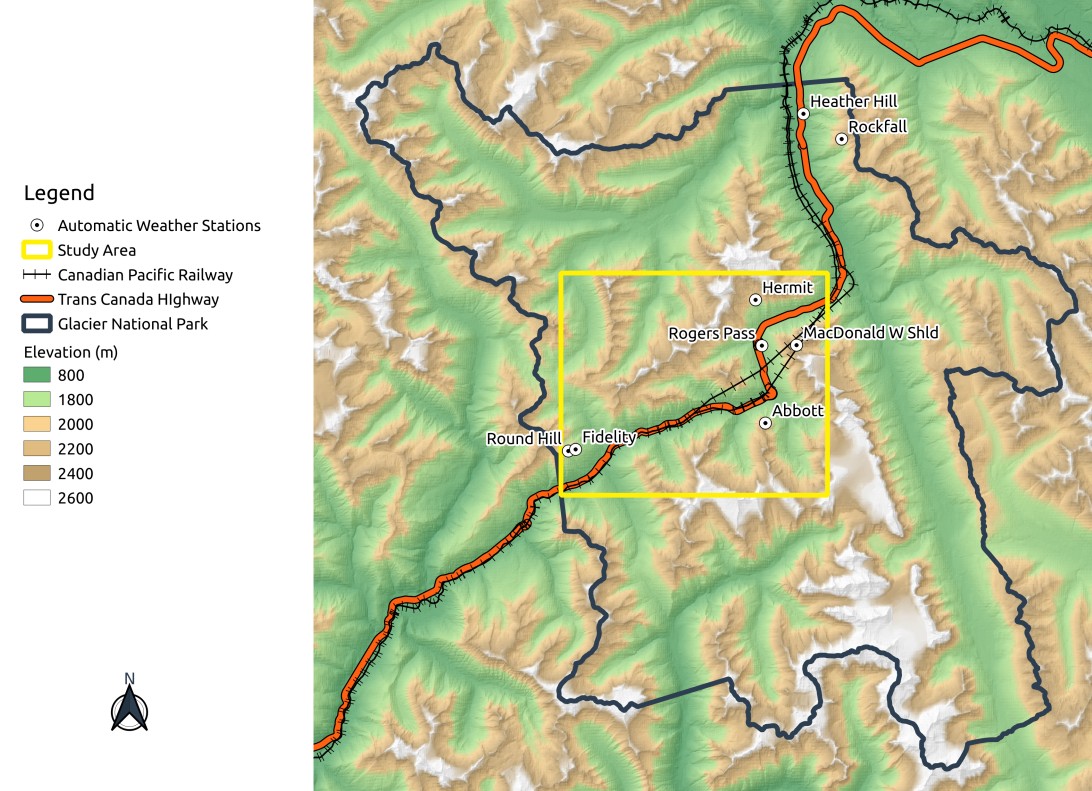

**Figure 1.** Glacier National Park, British Columbia, Canada

| Weather station | Elevation | TA | RH | VW | DW | PSUM | ILWR | ISWR | HS |
|---|---|---|---|---|---|---|---|---|---|
| Abbott | 2085 m | X | X | X | X | X | | | X |
| Hermit | 1950 m | X | X | X | X | X | | | X |
| Fidelity | 1905 m | X | X | X | X | X | X | X | X |
| McDonald W shoulder | 1930 m | X | X | X | X | | | | |
| Rogers Pass | 1315 m | X | X | X | X | X | | | X |
| Round Hill | 2100 m | X | X | X | X | X | | | |

**Table 1.** Inventory of instruments present on the study site weather stations. TA stands for Air Temperature, RH for Relative Humidity, VW for Velocity of Wind, DW for Direction of Wind, PSUM for Precipitation water equivalent, ILWR for Incoming Long Wave Radiation, and ISWR for Incoming Short Wave Radiation

## 3 The Numerical Weather Predictions downscaling processing chain design

Figure 2 summarizes the Numerical Weather Predictions (NWP) downscaling processing chain.

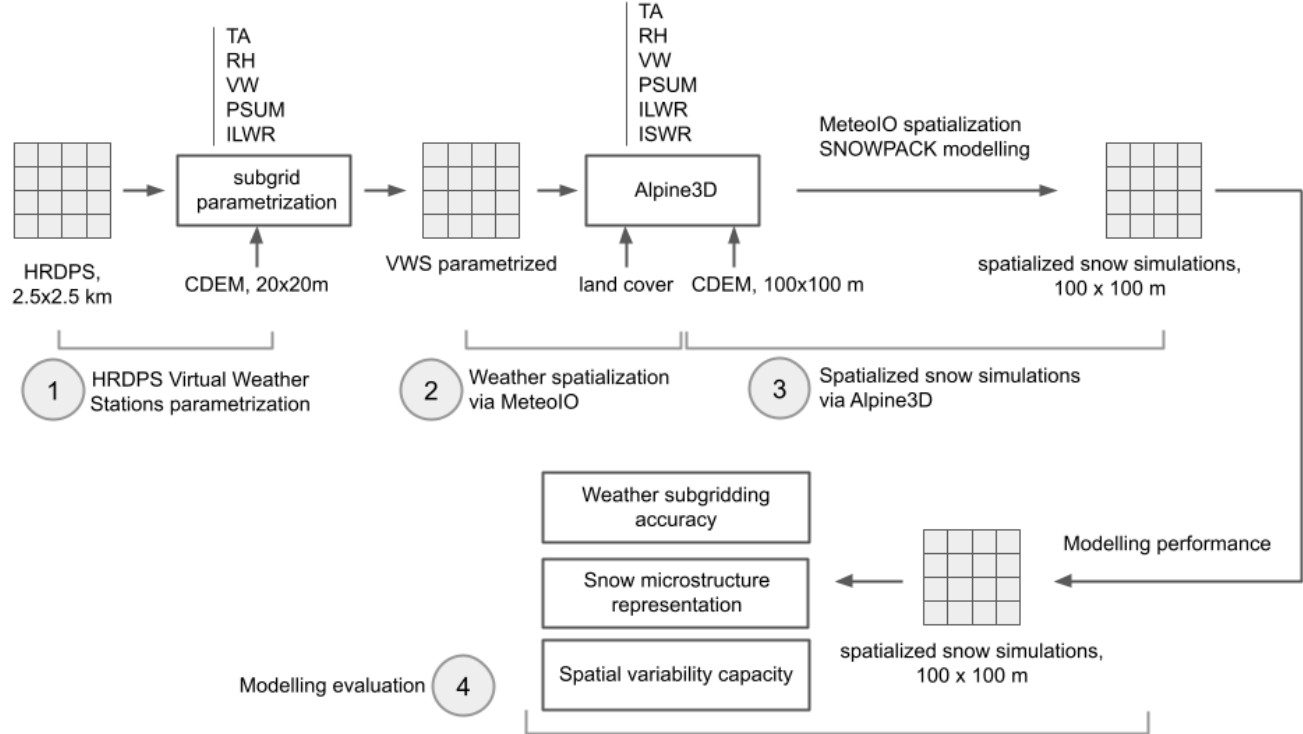

**Figure 2.** Numerical Weather Predictions downscaling scheme flowchart

## 3.1 HRDPS subgridding and Alpine3D simulations

The High Resolution Deterministic Prediction System (HRDPS) produced by the Meteorological Service of Canada provides a 2.5 km gridded hourly forecast of atmospheric variables for most of Canada (Milbrandt et al., 2016). Atmospheric variables are computed for each pixel at a reference elevation provided by the underlying 2.5 km resolution Digital Elevation Model. This study is based on a grid composed of 70 HRDPS prediction cells overlying the study area, and including the following variables: TA, RH, VW, DW, PSUM, ISWR, and ILWR. First, using the 20 m Canadian Digital Elevation Model (CDEM), we transformed each HRDPS cell data into a Virtual Weather Station. To do so, each cell centroid coordinates was recomputed, minimizing the hypotenuse distance between each underlying CDEM pixel centroid and the HRDPS centroid, in an 800 m radius around the original HRDPS centroid. Then, TA, RH, PSUM and ILWR were parameterized to correct for the model's biases and account for the elevation discrepancy between the HRDPS cell elevation and the new centroid CDEM elevation. VW was parameterized to account for the topography underlying the 2.5 km resolution grid unresolved by HRDPS.

Using all weather stations in the Park, bias in air temperature was found to have a non-linear relationship with the elevation difference between the station elevation and the original HRDPS cell elevation over the 2018-2020 period. A training set was generated by randomly selecting 75% of this dataset uniformly across elevations, and the remaining 25% served as validation

set. The data was transposed into logarithmic space to perform a linear regression. The resulting logarithmic fit was then applied over the TA dataset when the elevation difference between the Virtual Weather Station and its overlying HRDPS cell was over 100 m.

$$TA_p = \begin{cases} TA_{hrdps} + ln(-0.012\Delta_E + 6.89) & \text{if } |\Delta_E| > 100 \\ TA_{hrdps} & \text{otherwise} \end{cases} \tag{1}$$

where:

$\Delta_E$ corresponds to the elevation difference between the CDEM cell and the HRDPS cell, $TA_p$ is the parameterized air temperature, and $TA_{hrdps}$ is the raw HRDPS air temperature.

RH was corrected by first converting relative humidity to dew point temperature, as described in Liston and Elder (2006). This dew point temperature was then adjusted using the logarithmic fit presented above, and converted back to relative humidity.

Snowfall was first parameterized using an elevation lapse-rate correction. This lapse rate was computed by performing a simple linear regression of precipitation as a function of elevation. We used a dataset of four weeks of manual SWE measurements on four conventional HN24 precipitation boards placed between 1330 m and 1920 m at Mt Fidelity, all placed in flat and open areas sheltered from the wind.

$$PSUM_p = PSUM_{hrdps} + 0.0011 \times \Delta_E \times PSUM_{hrdps} \tag{2}$$

Finally , the HRDPS ILWR was downscaled using the lapse-rate correction developped by Marty et al. (2002), and VW was downscaled to the 20 m CDEM resolution at each new centroid position using the Sky View Factor approach (Helbig and Löwe, 2014; Helbig et al., 2017).

These Virtual Weather Stations were then spatially interpolated on a 100 m grid via MeteoIO (Bavay and Egger, 2014) using the CDEM grid resampled to 100 m. TA was spatialized using a simple lapse rate computed from the AWS data and Inverse Distance Weighting (IDW). RH, VW, and DW were spatialized using the Micromet algorithms described in Liston and Elder (2006). To spatialize precipitations, we used topographic parameters and prevailing winds to alter the precipitation field, to account for wind snow redistribution (Winstral et al., 2002). Incoming shortwaves were spatialized considering terrain shading, slopes and reflections from neighboring cells. Finally, ILWR was spatialized using IDW. All the spatial interpolation algorithms mentioned above are a part of the MeteoIO library, which is integrated into the Alpine3D model.

Alpine3D is a spatially distributed 3D model, which allows running the vertical 1D snow model SNOWPACK over an area while taking into account the spatial processes affecting affecting the input atmospheric variables, such as terrain shadowing (Lehning et al., 2006). SNOWPACK is a detailed multi-layer thermodynamic finite-element model of snow microstructure and metamorphism. In this model, the snow microstructure is represented by four main variables: grain size, bond size, dendricity and sphericity for each snow layer. In addition, the model simulates several metrics of interest when monitoring the evolution of the snowpack, such as height of snow, SWE, density, optical grain size, or snow temperature (Bartelt and Lehning, 2002; Lehning et al., 2002). To do so, the model is fed with three text files describing the weather parameters on the time domain of the simulation, the initial state of the soil layers on which the snow is going to develop (and initial snow layers if relevant),

and finally the configuration of the simulation. The model was run at Rogers Pass on the same 100 m grid described above, over an area of 18 km × 16 km (288 km$^2$) centered on the Highway 1 corridor for winters 2018–2019 and 2019–2020. The snowdrift scheme was turned off, and we generated outputs for three reference stations: Fidelity, Hermit, and Abbott. To assess the spatial variability capacity of the subgridding framework, the model was ran on the hole simulation domain, and we also generated outputs at six points within the same cell for intra-cell variability assessment. The specific cell was chosen because it is the only cell in the simulation domain that features a north and south slope with elevations ranging from below treeline to the alpine on both aspects. No glacier is present in the area. Table 2 summarizes the topographic characteristics for the chosen intra-cell spatial variability points.

| Spatial variability Point | Elevation (m) | Slope azimuth (°) | Slope angle (°) |
|---|---|---|---|
| S_BTL | 1510 | 144 | 17 |
| S_TL | 1871 | 140 | 23 |
| S_ALP | 2197 | 179 | 22 |
| N_BTL | 1548 | 359 | 25 |
| N_TL | 1852 | 356 | 26 |
| N_ALP | 2079 | 352 | 31 |

**Table 2.** Summary of the topographic characteristics for the chosen intra-cell spatial variability points. BTL stands for Below TreeLine, TL for TreeLine, ALP for Alpine

### 3.2 Validation data and atmospheric parameters subgridding evaluation

To compare with snow simulations driven by the raw HRDPS and the subgridding framework, we performed SNOWPACK simulations driven with AWS data at Fidelity, Hermit and Abbott stations. We filtered weather station data to remove outliers; data gaps smaller than six hours were linearly interpolated while larger gaps were filled using parameterized HRDPS data. Note that the Abbott station is located in a thunderstorms-prone area. Hence, the station is shut down all summer and is only turned back on mid-October. Parameterized HRDPS data are again used to fill in this gap. Moreover, SR50 snow depths measurements at Fidelity and Abbott stations were compared against each snow simulation approach.

The Numerical Weather Forecast subgridding quality was statistically assessed using three well-known criteria: bias, Mean Absolute Error (MAE), and Spearman R correlation coefficient. These indicators allowed to quantify respectively the systematic difference between the models and the ground-truth measurements at the AWS, the prediction accuracy, and the strength of the association between modeled variables and the ground truth. To smooth small time lags between modeled meteorological events and measurements at the AWS, we averaged the meteorological time series over 2-hour time steps, and reaccumulated precipitations over the same period.

### 3.3 Snow modeling evaluation

Dynamic Time Warping (DTW) is an algorithm developed to measure the similarity between two sequences. In a nutshell, DTW computes the optimal match between two signals, while allowing for an elasticity in time (or space, in the case of snow profiles). It first resamples the two sequences on 1D-grids of the same elemental size and length. Then, a local cost matrix $D$ is built, summarizing the distance between every elemental pair. From there, an accumulated cost matrix $G$ is built by computing the accumulated cost to iterate from one element of $D$ to the next one, respecting a predefined constraint set. The optimal alignment is found by minimizing the alignment accumulated cost.

Although originally designed for speech recognition (Sakoe and Chiba, 1978), DTW is extensively used in time series analysis, and it has recently received an increased interest by the snow community (Hagenmuller and Pilloix, 2016; Hagenmuller et al., 2018; Viallon-Galinier et al., 2020; Herla et al., 2021). In the snow science community, DTW has only been used so far in an avalanche forecasting perspective, focusing on aligning standard snow parameters (e.g., grain type, hardness, Liquid Water Content). In this study, we present a new development to the open-source DTW snow profile alignment package written by Herla et al. (2021), allowing to align snow profiles on remote sensing-oriented snow parameters, namely, layer density and Optical Grain Size (OGS), two key parameters in snow radiative transfer modeling. To do so, an alternative cost function was added to compute the local cost matrix $D$.

$$D_{i,j} = w_d d_d(q_i^d, r_j^d) + w_{ogs} d_{ogs}(q_i^{ogs}, r_j^{ogs}) \tag{3}$$

where $w_d$ and $w_{ogs}$ are averaging weights respectively applied to density and OGS ($w_d + w_{ogs} = 1$), $r_n^k$ denotes the $n^{th}$ element of the reference profile $R$, and $q_n^k$ denotes the $n^{th}$ element of the query profile $Q$. Finally, $d_d()$ and $d_{ogs}()$ correspond to the distance function for density and OGS respectively, which is simply the absolute difference between the two elements, normalized over the entire vertical profile.

The space elasticity in the alignment algorithm allows to find the best match for a layer of the query profile in the same depth range in the reference profile. The algorithm constraints define the amount of elasticity allowed, i.e., the warping window definition and the local slope constraint. These constraints are essential to keep the algorithm from degenerating and generate irrelevant alignments. However, HRDPS tend to underestimate precipitations in mountain environments (Bellaire et al., 2011, 2013; Côté et al., 2017). Hence, profiles generated from atmospheric models and ground truth profiles can have a significant difference in HS. Physically matching layers are then too far apart, according to the algorithm's constraints, preventing the algorithm from generating relevant matches. Therefore, we artificially inflated profiles modeled using NWP, each layer being multiplied by the height ratio with the station profile. This allowed to rely solely on snow microstructure parameters for the alignment, assessing only the microstructure representation. The generated aligned (or warped) profile was then used to compute a mean bias for density and OGS with respect to the ground truth. As precipitation is usually underestimated by HRDPS, HS should be underestimated as well, which should impact the overburden pressure on basal layers. This might result in a small negative bias on density with regards to AWS driven SNOWPACK runs, depending on the amount of missing snow. For OGS, the temperature gradient in this region is low and metamorphism mainly happens through gravitational settling, leading to little variability in OGS in the snowpack (Madore et al., 2018). As a result, we do not expect much impact of the inflation

approach on this microstructure parameter, as the main discrepancies should come from offsets in rain-on-snow modeling, and melt/percolation events. Height of Snow was compared to the station's SR50 measurements when available, and the Nash-Sutcliffe model efficiency coefficient (Nash and Sutcliffe, 1970) allowed to assess the SWE modeling quality using HRDPS and subgridded HRDPS data versus the station runs over the two seasons. For more details on the DTW implementation used in this paper, please refer to Herla et al. (2021).

## 4   Results

### 4.1   Numerical Weather Forecast subgridding performance

Figure 3 summarizes the performances of the subgridding framework (denoted as SGF in the figures) applied to the 2018–2019 and 2019–2020 HRDPS time series. The SGF delivers a mixed performance for TA in 2018–2019. The HRDPS model shows a 1.8 °C negative bias at Abbott, which is reduced by 1.2 °C using the subgridding framework. However, the negative bias is increased by 0.6 °C at the Fidelity station and very slightly at Hermit station (<0.1 °C). On the other hand, the Mean Absolute Error (MAE) and the Spearman R coefficient are slightly increased for most of the validation stations. TA shows a very strong correlation with station measures (R>0.8).

Relative Humidity subgridding yields good performances as the HRDPS model bias and MAE are reduced by 1% to 6% at all sites. The bias is constant at Fidelity station, as is the MAE at Abbott. The Spearman correlation coefficient is also slightly increased at Hermit and Fidelity stations and only shows a slight decrease at Abbott. Finally, modeled RH shows a strong correlation with station values (0.6 <R <0.8).

Overall, the SGF performs best at subgridding VW. HRDPS is constantly overestimating wind speed by 1.5 m.s$^{-1}$. This bias is considerably reduced to 0.25 m.s$^{-1}$ on average, and MAE is reduced by around 1 m.s$^{-1}$. However, the correlation with station values is overall weak (0.2 <R <0.4), and the subgridding workflow seems to have even weakened this correlation, except for Hermit station, which shows a negligible correlation (R <0.2).

Finally, subgridding shows good performance for PSUM as well. In agreement with the literature, the bias in modeled precipitation shows a general lack of precipitation in the HRDPS model, and ranged from 0.05 mm to 0.2 mm. In 2018-2019, MAE values range between 0.25 mm and 0.35 mm. With the subgridding, bias and MAE decrease at Abbott, increase slightly at Hermit, and bias decreases at Fidelity while the MAE remained constant. Correlation with station values is moderate at Abbott and Hermit (0.4 <R <0.6) and strong at Fidelity (0.6 <R <0.8). Finally, subgridding slightly improves the correlation with AWS measurements at the former sites and stayed constant at the latter. For 2019–2020, the results are very similar to the 2018–2019 season; the bias and MAE are corrected on the same scale and the Pearson R coefficient is on the same range for each variable. The only notable difference is that the PSUM bias at Abbott is positive for this season, meaning that HRDPS overestimated precipitations, which is highly unusual. As a result, the SGF introduces even more precipitation bias (+ 0.07 mm).

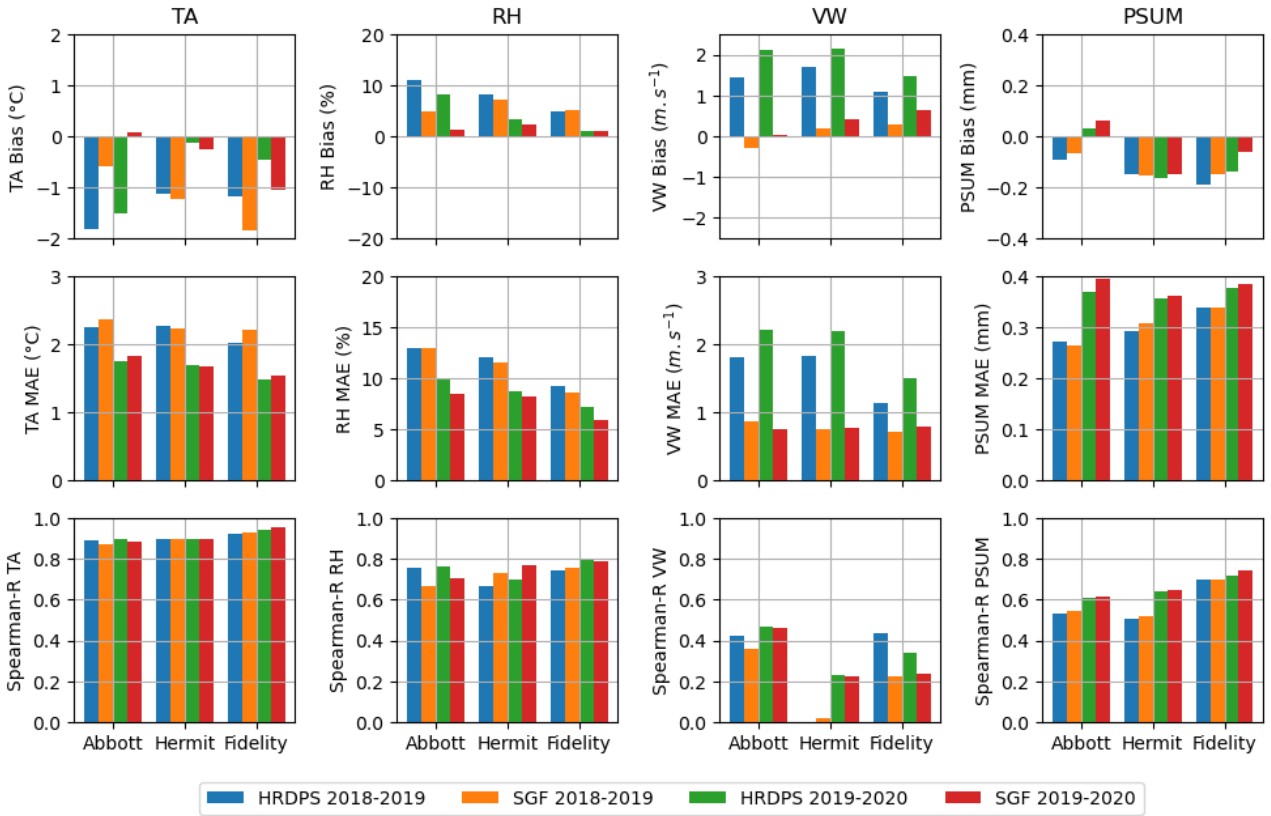

**Figure 3.** Atmospheric parameters evaluation for seasons 2018–2019 and 2019–2020. The first row shows biases with respect to AWS measurements for each atmospheric parameter, the second row shows Mean Absolute Error, and the third shows Spearman-R coefficients. Blue and orange bars refer to values for 2018-2019 raw HRDPS and subgridding framework respectively. Green and red bars refer to values for the 2019-2020 season, in the same order.

## 4.2 Subgridding performance for snow modeling

Figures 4 and 5 summarize the subgridding framework performances for seasons 2018–2019 and 2019–2020. From here, snow simulations are denoted as SGF-SNOWPACK, HRDPS-SNOWPACK, and AWS-SNOWPACK when driven respectively by subgridded atmospheric parameters from the SGF, raw HRDPS forecasts, and AWS measurements. For 2018–2019, snow-pack similarity shows the same behaviour at every site and for both HRDPS-SNOWPACK and SGF-SNOWPACK. The season begins with average similarity values (around 0.5), then it plummets to low values in mid-October (<0.5) before improving to higher levels of similarity in November and for the rest of the season (0.6 <sim <0.8). In general, HRDPS-SNOWPACK tend to have a closer similarity with AWS-SNOWPACK early in the season. SGF-SNOWPACK tend to score higher similarities than HRDPS-SNOWPACK in the mid-season before converging back with HRDPS-SNOWPACK in the spring. For the 2019–2020 season, similarity is again highly variable around 0.5 at Abbott and Hermit for both HRDPS-SNOWPACK and

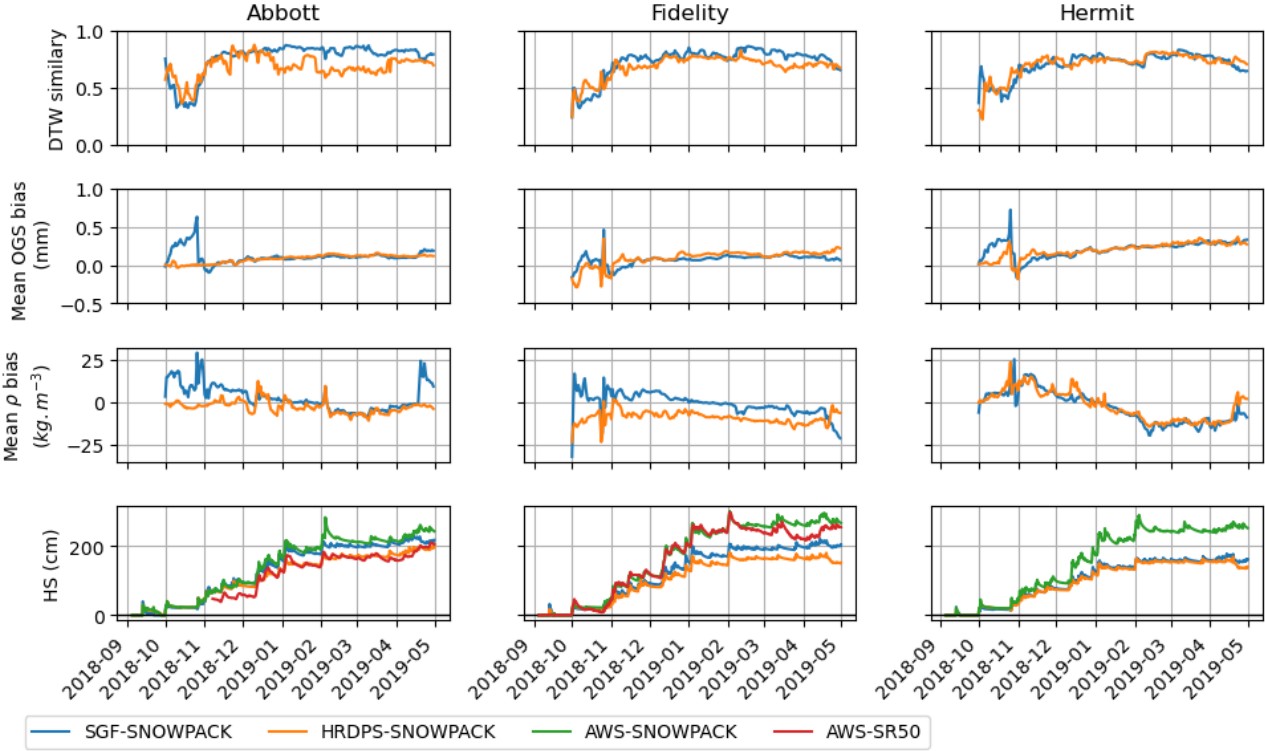

**Figure 4.** Snowpack similarity assessment for season 2018–2019. The first row shows DTW similarity for each station, the second shows bias in OGS with respect AWS-SNOWPACK, the third shows density bias with respect to AWS-SNOWPACK, and the last row shows simulated HS (and measured when available). The blue curves refer to SGF-SNOWPACK, the orange curves to HRDPS-SNOWPACK, and the green curves to AWS-SNOWPACK. In the HS plots for Abbott and Fidelity, the red curves refer to SR50 measurements at the station plot.

SGF-SNOWPACK. The early season at Fidelity shows very low similarities for SGF-SNOWPACK and HRDPS-SNOWPACK. Then, starting in November, the similarities stabilize and slowly rise throughout the season to 0.8 at every site. Again, SGF-SNOWPACK shows a higher similarity than HRDPS-SNOWPACK during the mid-winter period at Abbott and Hermit. HRDPS-SNOWPACK reaches the same level of similarity by early spring or the end of the winter, respectively. At Abbott,
HRDPS-SNOWPACK and SGF-SNOWPACK similarities are very similar, though the former shows more fluctuations during most of the winter and early spring. The average similarity for all sites and seasons is 0.8 for SGF-SNOWPACK and 0.75 for the HRDPS-SNOWPACK. Overall, SGF-SNOWPACK increased snowpack similarity by 7% at Abbott, by 2% at Hermit, and by 6% at Fidelity with respect to HRDPS-SNOWPACK and when compared against AWS-SNOWPACK. The mean error in OGS shows a similar behaviour to similarity at every site in 2018–2019 for both approaches. The error peaks and fluctuates
strongly in the early season for both approaches, then stabilizes around mid-November. Considering the high variability in the fall for both seasons (September to November included), the first three months were considered as a spin-up phase for the model to initiate a proper snowpack. Hence, the numerical analysis of the results was carried out starting on the first of December.

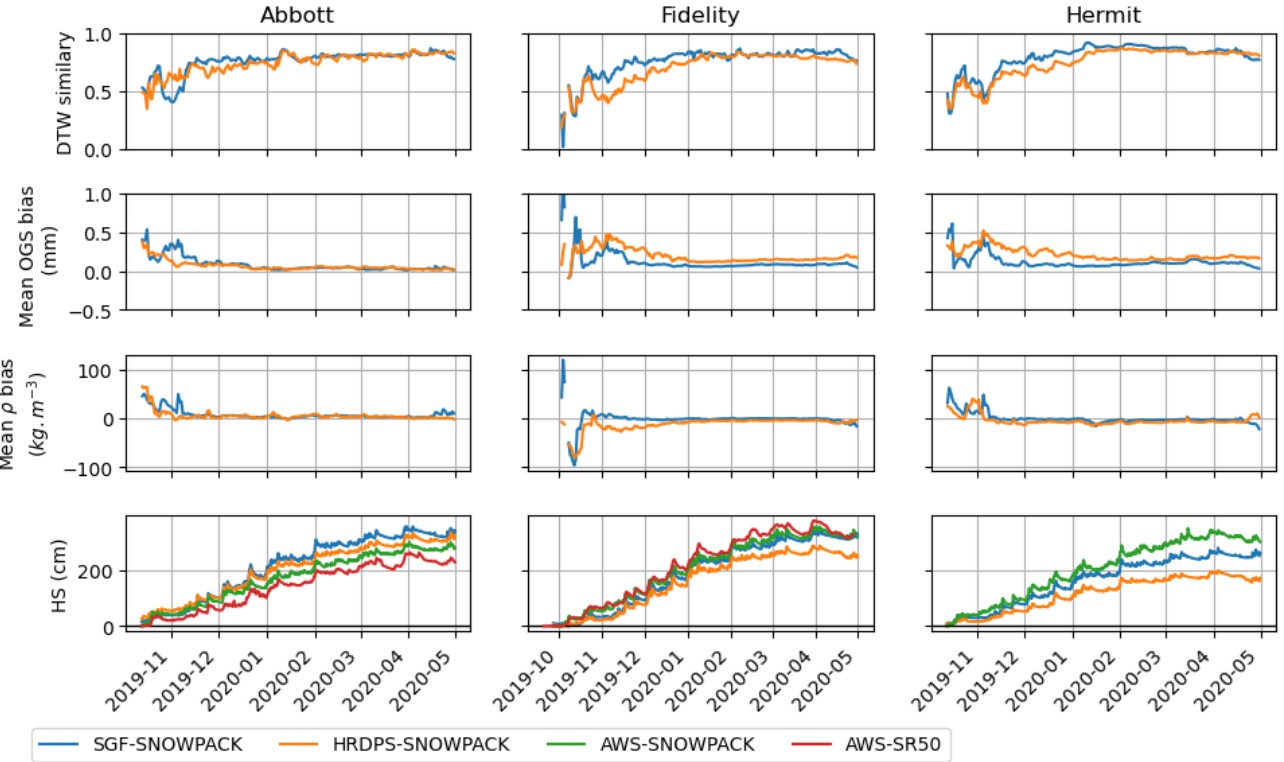

**Figure 5.** Snowpack similarity assessment for season 2019-2020. The first row shows DTW similarity for each station, the second shows bias in OGS with respect AWS-SNOWPACK, the third shows density bias with respect to AWS-SNOWPACK, and the last row shows simulated HS (and measured when available). The blue curves refer to SGF-SNOWPACK, the orange curves to HRDPS-SNOWPACK, and the green curves to AWS-SNOWPACK. In the HS plots for Abbott and Fidelity, the red curves refer to SR50 measurements at the station plot.

Both HRDPS-SNOWPACK and SGF-SNOWPACK are slightly overestimating OGS with respect to AWS-SNOWPACK, but overall, the proposed method allows decreasing the bias by 0.04 mm at Fidelity. However, OGS bias remained constant at Ab-
bott and Hermit on the same period. The same pattern repeats for the 2019–2020 season at all sites, where SGF-SNOWPACK decreases on average the OGS bias by 0.07 mm at Fidelity and 0.09 mm at Hermit. The bias in OGS remained unchanged at Abbott once again this season.

Similarly to OGS, in 2018–2019, the mean error in density shows higher values in the early season, and stronger fluctuations for both HRDPS-SNOWPACK and SGF-SNOWPACK. Variations tend to stabilize by mid-November, and the error increases
again towards the end of the season. Again, the numerical analysis was performed from the first of December until the end of the simulation for both seasons. Generally, HRDPS-SNOWPACK seems to underestimate snow density. SGF-SNOWPACK brought density 3.17 kg.m$^{-3}$ closer to AWS-SNOWPACK at Abbot and a 6.79 kg.m$^{-3}$ closer at Fidelity for the 2018–2019 season. However, on the same period at Hermit, the density bias increased by 1.71 kg.m$^{-3}$ on average. In 2019–2020, the density bias decreased on average by 4.65 kg.m$^{-3}$ at Fidelity and 3.62 kg.m$^{-3}$ at Hermit. However, the density bias increased

by 0.9 kg.m$^{-3}$ over the same period at Abbott.

Finally, SGF-SNOWPACK mean error on modeled HS is 35 cm in 2018-2019 (20 cm improvement when compared to HRDPS-SNOWPACK), and 29 cm in 2019–2020 (29 cm improvement) when compared with the SR-50 measurement at Fidelity. For reference, the modelled HS with AWS-SNOWPACK shows a mean error of 8 cm in 2018-2019 and 14 cm in 2019-2020. However, SGF-SNOWPACK seem to degrade the quality of the HS modelling with regards to HRDPS-SNOWPACK when

compared to the SR50 at Abbot station, overestimating HS for each season. Yet, the Station-SNOWPACK run at Abbott shows a high discrepancy with the SR-50 measurements as well, overestimating HS as well (especially in 2018-2019). Finally, HS remained relatively unchanged at Hermit for 2018–2019 as the framework did not bring a substantial improvement when compared to the Station-SNOWPACK modelled HS.

Overall SWE modeling was considerably improved at all stations except at Abbott in 2019–2020 (Figure 6) when compared

to AWS-SNOWPACK. Table 3 summarizes the Nash Sutcliffe Efficiency coefficient (NSE) values at each site and for each season. On average, SGF-SNOWPACK improves the SWE NSE by 13%, up to 57% at Hermit in 2019-2020.

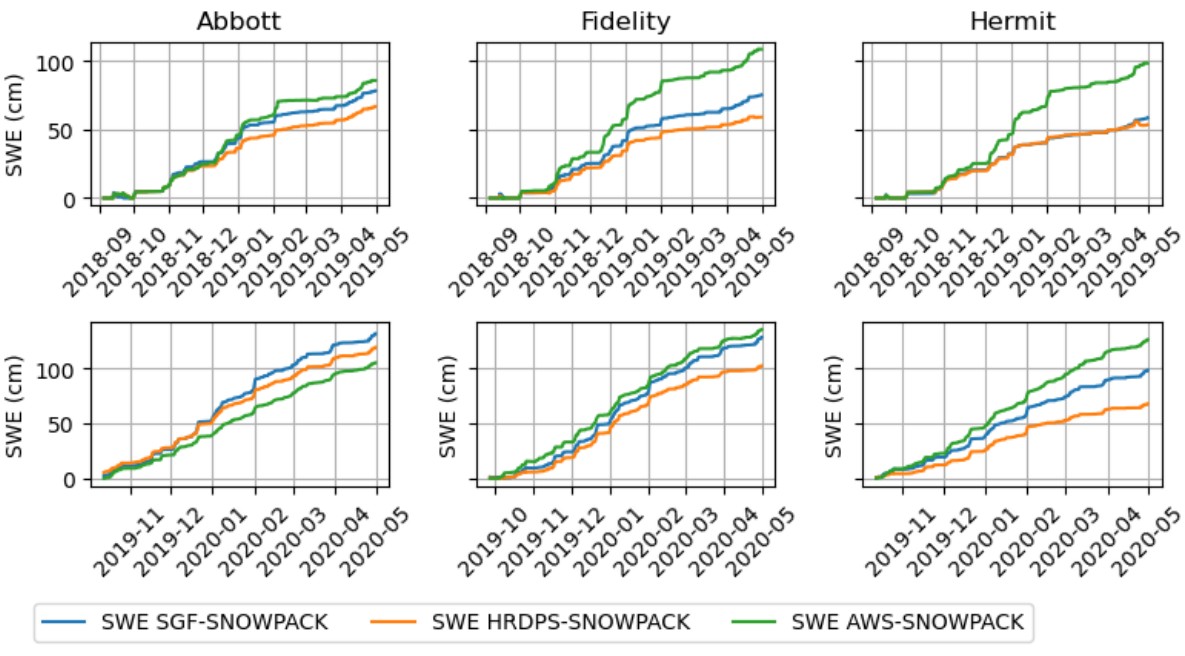

**Figure 6.** SWE modeling for seasons 2018–2019 and 2019–2020. The first row represents SWE simulations at each site for the 2018-2019 season, and the second row represents SWE simulations for the 2019-2020 season. The blue curves refer to SGF-SNOWPACK, the orange curves refer to HRDPS-SNOWPACK, and the green curves refer to AWS-SNOWPACK.

| | Abbott | | Fidelity | | Hermit | |
|---|---|---|---|---|---|---|
| | 2018-2019 | 2019–2020 | 2018-2019 | 2019–2020 | 2018-2019 | 2019–2020 |
| SGF-SNOWPACK | **0.97** | 0.64 | **0.72** | **0.81** | **0.53** | **0.85** |
| HRDPS-SNOWPACK | 0.82 | **0.86** | 0.43 | 0.77 | 0.51 | 0.28 |

**Table 3.** Nash-Sutcliffe model Efficiency coefficient for SWE at each site for each season

### 4.3 Intra-cell spatial variability provided by the subgridding framework: a case study

Figure 7 summarizes the main atmospheric parameters values for the six spatial variability sites, averaged per month (and accumulated for precipitations). First, the lapse-rate applied for TA downscaling and spatialization respects the general rule of thumb that TA should get colder with elevation. However, the selected point on the North aspect is a 100 m lower than the South aspect point, and as a result, the figure shows slightly warmer temperatures consistently throughout the season on the North aspect. Second, the aspect gradient is respected with lower incoming shortwave radiations and slightly lower temperatures in the north aspects. Moreover, wind direction is mostly coming from the South, South-West. South slopes are thus more exposed to the wind and north aspects are more sheltered. This is reflected by wind speed values being higher in the south aspects, especially in the alpine. Snow redistribution by the wind is accounted for, leeward slopes getting more snow than windward slopes. Finally, the altitudinal precipitation rate gradient is also respected by the subgridding framework, with precipitation rates getting higher with elevation. This atmospheric parameters variability is then propagated to the modeled snowpacks (Figure 8). Again, the altitudinal gradient in HS is present, with a deeper snowpack from below treeline to the alpine. The wind effect on the snowpack is also well represented in the simulations. Indeed, dominant winds are blowing from the South / South-West, and as a result southern slopes are affected by stronger winds (Figure 8). In the SNOWPACK model, grain type is a function of dendricity and sphericity, two parameters governed by the temperature gradient within the snowpack. As the surface temperature is altered by surface winds, precipitation particles (lime green) on the south aspects tend to metamorphose faster into decomposing and fragmented precipitation particles (dark green) than in the northern aspects, especially in the alpine. Finally, the melt onset date is a few days earlier on the south aspect than on the north aspect, and water is percolating faster and deeper in the snowpack on the south aspects.

Season 2019–2020 results for spatial variability appear in Appendix A, Figures A1 and B1. The subgridding framework creates the same altitudinal and aspect gradients, with more precipitations overall, milder temperatures, and stronger wind speeds on average.

### 4.4 Spatial variability of Snow Water Equivalent provided by the framework over the simulation domain

Figure 9 shows the evolution of simulated SWE averaged by elevation band and aspect. The elevation gradient is well represented over all four quadrants. In the East and West aspects, which corresponds to dominant winds in the area, the High alpine elevation is showing equal or lower SWE that in the alpine band. This reflects the effect of wind on ridges, modelled in the

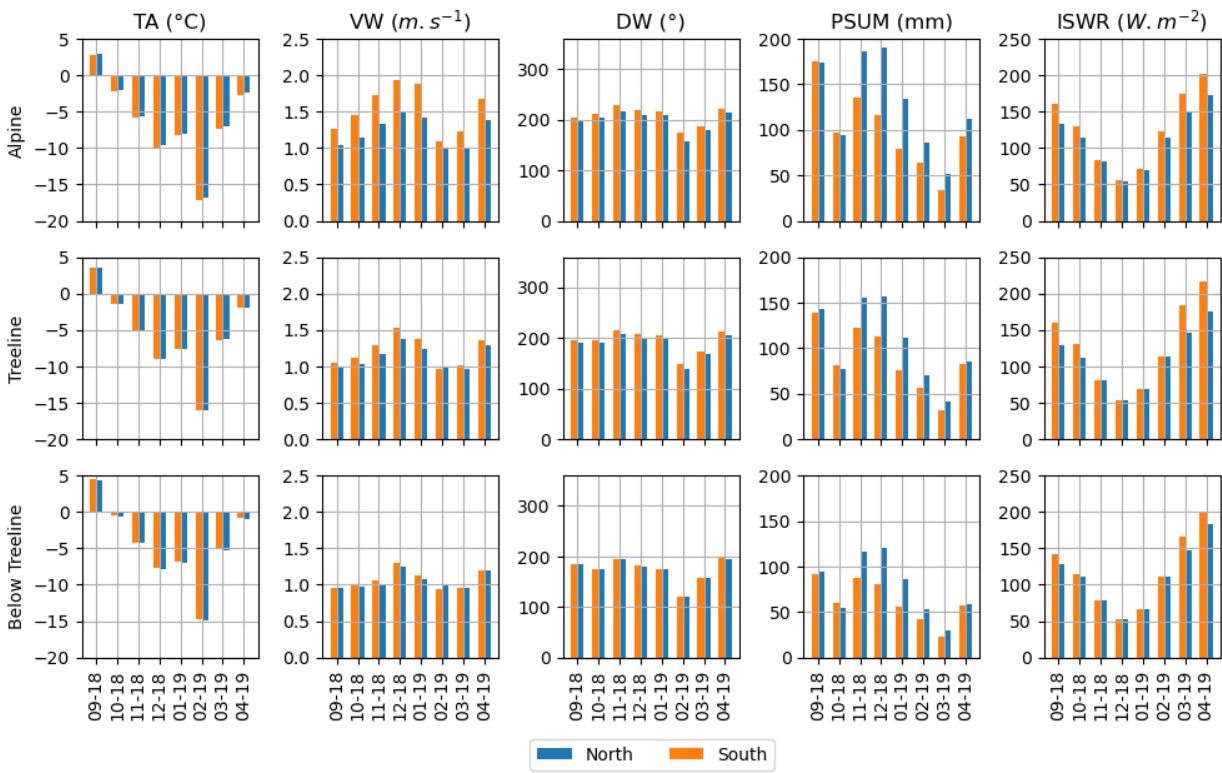

**Figure 7.** Intra-cell spatial variability of subgridded atmospheric parameters for season 2018–2019. Each row represents monthly averages for each atmospheric variable and cumulative precipitations, for each elevation band. Blue bars correspond to the North aspect, orange bars correspond to the South aspect.

SGF as an alteration of the PSUM field with wind speed and terrain features. Figure 10 shows the variability of modelled SWE within each HRDPS cell over the simulation domain. The top plot shows the variability in the early season (2018-11-19), and the bottom plot shows the variability in the end of the season, when the snowpack is at its peak (2019-03-04). Each cell shows a wide spread of SWE values, which indicates that the subgridding of weather parameters is very effective in bringing spatial variability within each cell.

## 5 Discussion

The proposed downscaling framework brings a noteworthy improvement to modeled atmospheric parameters when compared to station values. Bias and MAE for VW and RH are constantly attenuated. For precipitation, in most cases, bias is corrected but MAE increases. Indeed, even though the model on average underestimates precipitation, major precipitation events are overestimated (Côté et al., 2017). The "one-directional" lapse-rate correction reduces the overall bias by accurately correcting the small and common underestimation errors but accentuates the overall larger and rarer overestimation errors, thus increasing

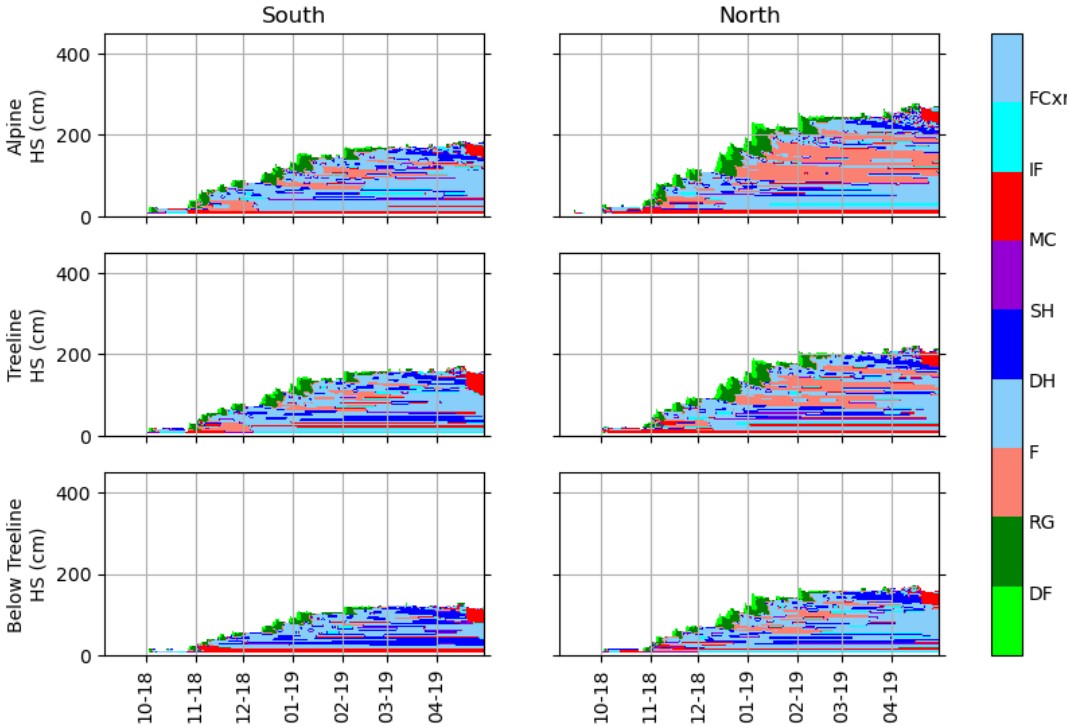

**Figure 8.** Snowpack spatial variability assessment for season 2018–2019. The left column represents snow profiles on the south aspect in the alpine, treeline, and below treeline elevation bands. The right column represents snow profiles at the same elevation bands, but on the north aspect. Colors represent grain type, which are denominated according to the international classification of Fierz et al. (2009)

the MAE. We acknowledge that it limits our work, and our lab currently carries out research to produce an adaptive bias
correction algorithm. Regarding TA, the mean systematic bias is slightly aggravated at every station except at Abbott, where bias is clearly improved. However, the added error remains under 0.5 °C. The same observation can be made about MAE, with the exception being at Hermit, where the MAE slightly lowers (i.e., improved precision). However, TA presents a positive bias at these 3 stations (i.e., HRDPS is colder than station measurements), which are all higher in elevation than the nominal elevation of their corresponding HRDPS cell. As a result, a naive Inverse Distance Weighting lapse-rate spatialization scheme
would have introduced even more bias and MAE in the system, aggravating the original error. The logarithmic bias correction reduces the bias in TA. However the IDW spatialization depends on the elevation difference between the HRDPS cell and each sub-pixel. As such, a positive TA bias at the parameterization stage is necessarily aggravated by IDW if the sub-pixel is higher in elevation. Thus, we then argue that the logarithmic lapse-rate parameterization allows reducing the error in the subgridding scheme and keeps it within a physically meaningful interval regarding spatial resolution.

Prior to discussing the accuracy of the subgridding framework for modelling snow properties, the use of a SNOWPACK run driven by AWS measurements as validation tool for the subgridding framework needs to be discussed. Madore et al.

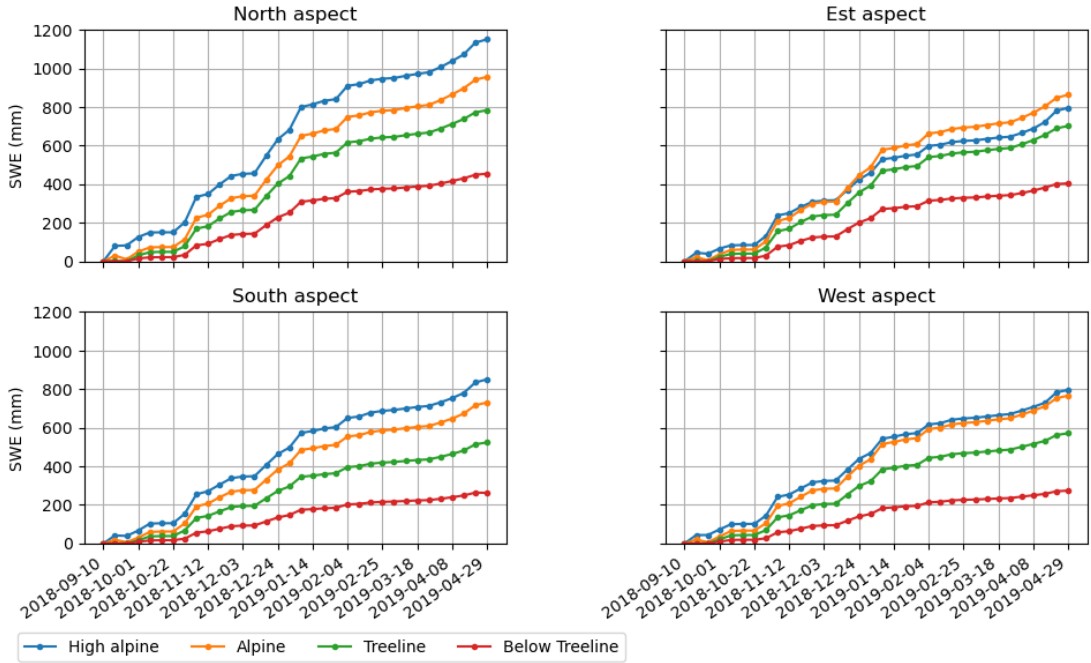

**Figure 9.** Average SWE aggregated by elevation band and aspect for season 2018–2019. Atmospheric parameters spatial variability assessment for season 2018–2019. Red line correspond to Below Tree Line elevations (< 1850 m), green is Tree Line (1850 m < elevation < 2000 m), orange is Alpine (2000 m < elevation < 2900 m), and blue is High Alpine (> 2900 m)

(2022) performed a detailed parameterization and analysis of the SNOWPACK model at Fidelity station. Results show that SNOWPACK performs very well on modelling HS and SWE, slightly underestimating both parameters ($-4.2\%$ mean error in bulk density). When performing a layer-by-layer comparison with observed snow profiles, SNOWPACK models density accurately with a slope of 0.88 and a correlation of 0.97, though low density snow was observed to be overestimated by the model, and high density snow was overestimated. With regards to OGS, Madore et al. (2018) showed that SNOWPACK usually overestimates OGS and introduces more variability in OGS than what is usually observed on the field. However, this lack of variability could be an artifact of the resolution of the sampler used in the field (5 cm). The accuracy of SNOWPACK at Fidelity has been proven and its biases are clear, therefore it seems more than reasonable to consider a SNOWPACK run driven by AWS as ground truth at Fidelity. However, the quality of a SNOWPACK run is almost entirely constrained by the quality of its inputs. Abbott and Hermit stations are more remote, harder to reach, and hence more difficult to maintain. As a result, it was not possible to conduct a similar study at these sites, and it is safe to assume that the AWS SNOWPACK runs present substantial bias with respect to reality. This is particularly well illustrated by the results on snow height modelling. When compared against the station SR-50 measurements, SGF-SNOWPACK drastically improves HS modelling at Fidelity over HRDPS-SNOWPACK for both seasons (especially in 2019-2020 where SGF-SNOWPACK is almost identical to AWS-SNOWPACK and very close to the SR50 measurements). However, at Abbott AWS-SNOWPACK is consistently overestimating HS with regards to the

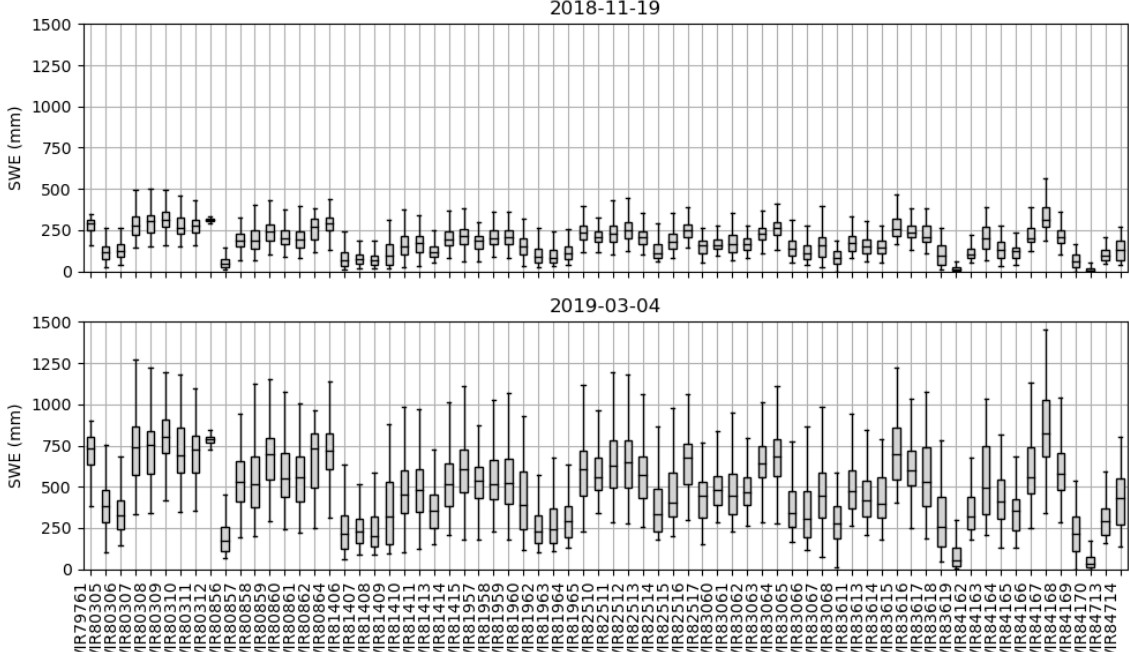

**Figure 10.** Boxplot of the SWE modelled by the subgridding framework within each HRDPS cell in the early season and in the end of the season. The box spans the interquartile range (IQR), the line represents the median, and the whiskers extend to the minimum and maximum value within 1.5 times the IQR. Outliers have been removed.

SR-50 measurements. The SGF is getting the SNOWPACK simulation closer to the AWS-SNOWPACK run, but actually further away from reality. Hence, even though the SGF improves weather parameters with respect to the measurements at the AWS, the weather inputs and the model parameterization fail to accurately represent reality at this site. This shows that weather measurements at Abbott are certainly dubious. Moreover, the SNOWPACK parameterization applied on every cell of the domain has been developed at Fidelity. The site is perfect for studying snow processes as it is sheltered from the wind, but it is not representative of the hole domain as it is quite idealized. Applying the same SNOWPACK parameterization on every cell of the domain can be misleading, and a specific parameterization should be applied as a function of topography.

Simulated snow parameters are usually improved, depending on the site and period. The most important errors and lowest similarities occur early in the season, when the snowpack is starting to build. This is due to differences in snow onset timing between simulations and reality, and milder air temperatures early in the season. As a result, there is a fine margin between solid (snow precipitation) and liquid (rain-on-snow), which can strongly alter the microstructure of the snowpack. Moreover, during this period, the snowpack is thinner, and discrepancies between layers have a heavier impact on the mean similarity and mean errors. With regards to microstruture, the subgridding framework allows decreasing by 0.06 mm the OGS overestimation compared with raw HRDPS simulations at Fidelity (averaged over both seasons), and by 0.07 mm at all sites and seasons (averaged over all sites and seasons). Optical diameter usually ranges from 0.1 mm to 0.4 mm in this region, and the average

snowpit OGS is 0.38 mm. Hence, the subgridding framework could improve the modelled OGS by 18% on average with respect to AWS-SNOWPACK modelled OGS. Furthermore SWE NSE is improved by 22.5% at Fidelity over all seasons, and by 16% when averaged over all sites and seasons with the AWS-SNOWPACK as a reference. The SWE bias is improved by 8.65 cm (both season average) at Fidelity, and by 5.67 cm on average over all sites and seasons. This represents a major improvement in a remote sensing perspective as SWE is the major driver for SWE inversion algorithms (King et al., 2015; Zhu et al., 2018; Tsang et al., 2022). For instance, King et al. (2015) reported a backscatter increase of 0.82 dB per 1 cm increase in SWE in the Ku-band with Canadian tundra measurements. Similarly, Yueh et al. (2009) found a 0.15 dB to 0.5 dB increase for every 1 cm increase in SWE at Ku-band in Colorado. According to these figures, using the subgridding framework in a SWE inversion context could provide an improvement of 1.4 dB to 7 dB in Ku-band simulated backscatter.

Finally, the subgridding framework performs well in introducing spatial variability over the simulation domain. SWE distribution across topographic categories respects the elevation gradient, the orientation of dominant winds in the area, and the erosion effect on ridges. Spatial variability is key when considering SAR signal inversion (King et al., 2018), and the subgridding framework should be a highly relevant tool in this context. However, the introduced spatial variability has not been evaluated against distributed snow height measurements, but airborne snow heights surveys are expensive and logistically challenging.

## 6  Conclusions

The work presented in this study aims at solving the present need in the snow remote sensing community for both the design and the evaluation of an atmospheric model subgridding framework to perform snow modelling in the context of coupling with a SAR signal inversion routine. To do so, (i) a new NWP downscaling approach was introduced, by first parameterizing the 2.5 km HRDPS cells into a Virtual Weather Station array, which was then spatially interpolated using the MeteoIO/Alpine3D models. (ii) Snow simulations were performed using the state-of-the-art model SNOWPACK. Microstructure modeling quality was assessed using the DTW algorithm and an original cost function focusing on density and optical grain size, and SWE modeling improvements were quantified using the Nash-Sutcliffe Efficiency coefficient. (iii) Spatial variability of atmospheric parameters and snowpack state variables within one subgridded HRDPS cell was assessed. Finally, the introduced spatial variability of SWE was assessed over the simulation domain, as well as intra-cell variability. The main conclusions of this study with respect to the research questions formulated in the introduction are:

1. How do subgridded HRDPS forecasts compare to reference Automatic Weather Stations in the simulation domain ?
   The atmospheric parameter subgridding framework yields an overall good performance, especially for RH and VW. However, research should be carried on to find a more suitable spatialization algorithm for air temperature, and an adaptative precipitation rate correction algorithm.

2. Do the resulting atmospheric forcings lead to an improvement in snowpack modelling, especially for critical snow parameters in remote sensing applications ?

- The general overestimation of OGS by SNOWPACK when driven by raw HRDPS data was decreased by 0.06 mm on average at Fidelity, and by 0.07 mm avergared over all sites. This represents a 18% improvement over raw HRDPS-SNOWPACK simulated OGS.

- The Nash-Sutcliff Efficieny coefficient for SWE was improved by 22.5% at Fidelity and 16% on average when compared to HRDPS-SNOWPACK simulations. SWE bias was diminished by 8.65 cm at Fidelity and 5.67 cm on average. This is a major improvement in a SAR remote sensing context, at this could lead to up to a 7 dB improvement in Ku-band simulated backscatter.

- In this context, the first three months (September to November) of snow simulations should be considered as a spin-up phase for the snow model, as discrepancies between reality and simulations are critical before the snowpack is properly established and the similarity stabilizes in December and onward.

3. Which degree of spatial variability is introduced by the subgridding framework with regards to snow parameters ?

The subgridding framework introduces a realistic spatial variability in the snowpack state variables, respecting altitudinal and orientation gradient as well as ridge effects. The framework brings substantial variability within each HRDPS, reflecting the high spatial variability of the snowpack at the kilometre range in a mountainous environment.

This study shows that downscaling NWP at a 100 m resolution can improve local representation of atmospheric values and, as a result, improve the snowpack state variables modeling and spatial variability of the snowpack in complex topography. Moreover, the modeling of the two key parameters for snow remote sensing, SWE and OGS, was improved. This work is highly relevant in a remote sensing context. The remote sensing community is currently pushing for new SAR satellite missions, and Observation System Synthetic Experiments have proven that satellite SWE measurements would substantially improve SWE products RMSE (Garnaud et al., 2019; Cho et al., 2023). This study provides finer spatial variability forcing data and improved simulated snowpack state variables regarding NWP driven simulations. As a result, snow simulations performed with such method can provide a realistic first guess estimate of the retrieval parameters, bringing a solid basis to overcome the non-unique solution issue in physical retrieval algorithms (Tsang et al., 2022) and steer away from empirical retrieval approaches. In this context, the next logical step is to design a SWE retrieval algorithm, taking advantage of the vast array of SAR satellites in orbit, such as Sentinel-1 (C-band), TerraSAR-X (X-band), or the SnowSAR mission concept (dual Ku-band) led by Environment and Climate Change Canada and the Canadian Space Agency (Derksen et al., 2021). Future work can thus focus on using these modeled snowpack state variables along with field inferred distributions as a basis for a SWE SAR retrieval algorithm.

*Code and data availability.* Code and installation guidelines for MeteoIO/SNOWPACK/ALPINE3D can be found at https://gitlabext.wsl.ch/snow-models. The snowpack DTW alignment package can be found at https://CRAN.R-project.org/package=sarp.snowprofile.alignment
Code for the subgridding framework and the data used in this study is available upon request.

*Author contributions.* PB conceptualized and led the research, wrote the subgridding framework code, did the formal analysis, and wrote the initial draft. AL and BM helped conceptualize the research, provided scientific inputs, and supervised the project. AL acquired the funding and the resources for the project. All authors reviewed and edited the paper.

*Competing interests.* Alexandre Langlois is a member of the editorial board of The Cryosphere.

*Acknowledgements.* This project was funded by the Search and Rescue New Initiatives Fund from Public Safety Canada (SAR-NIF), the Natural Sciences and Engineering Research Council of Canada (NSERC) and the Quebec Research Funds - Nature and Technologies (FRQNT). The authors would like to thank Jeff Goodrich and the Mount Revelstoke and Glacier National Parks staff for their support. The authors also thank Simon Horton and Florian Herla at Simon Fraser University for respectively providing HRDPS data in the .smet format and for onboarding PB in the snowpack DTW package. Finally, the authors thank Nora Helbig and Mathias Bavay for their advice and help with the Alpine3D model.

445

450

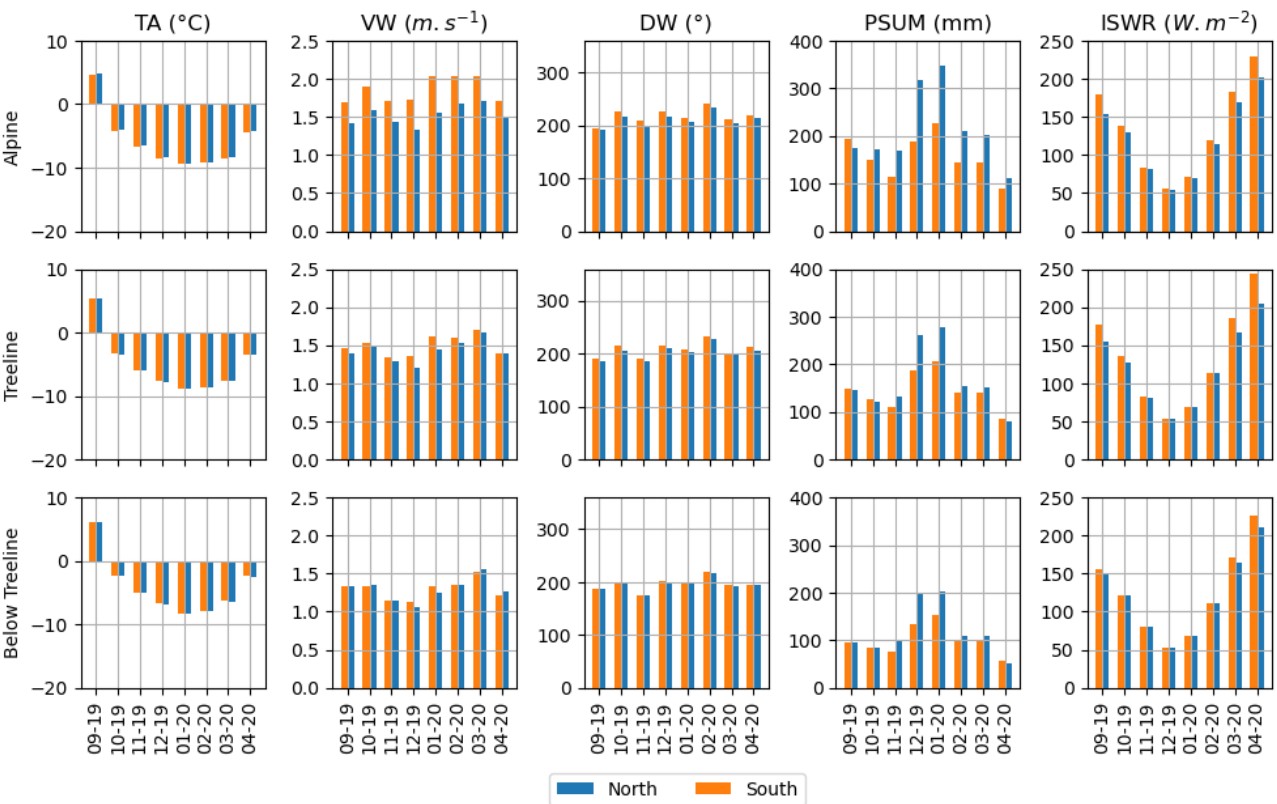

**Figure A1.** Intra-cell spatial variability of subgridded atmospheric parameters for season 2019-2020. Each row represents monthly averages for each atmospheric variable and cumulative precipitations, for each elevation band. Blue bars correspond to the North aspect, orange bars correspond to the South aspect.

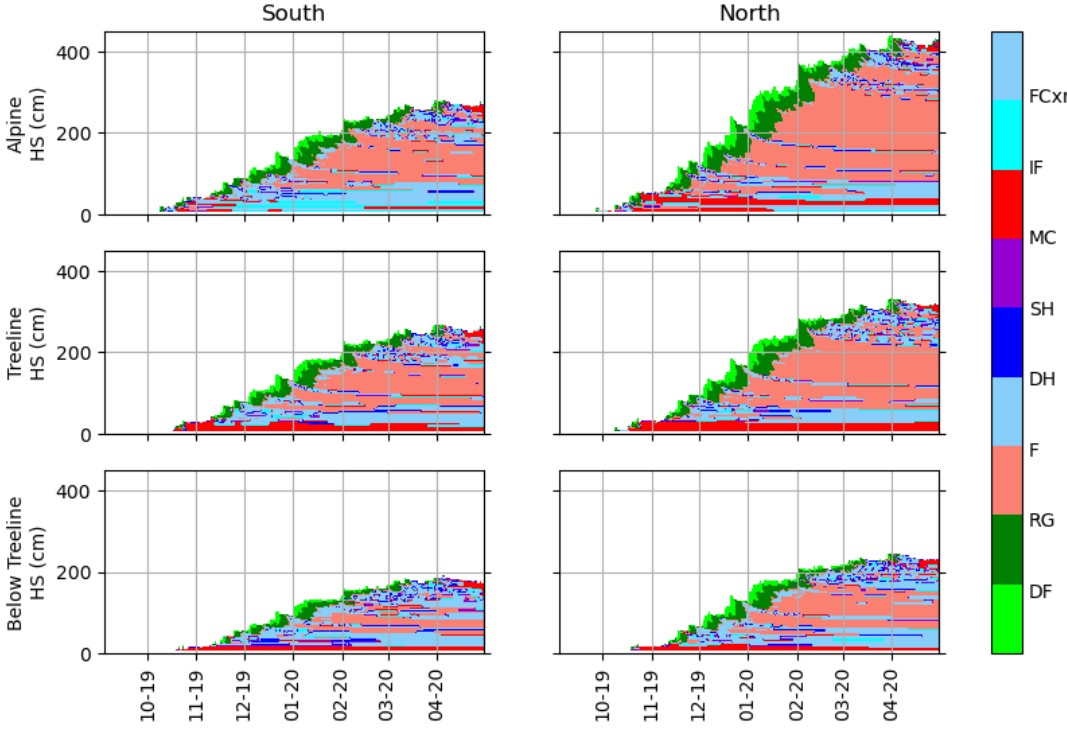

**Figure B1.** Snowpack spatial variability assessment for season 2019-2020. The left column represents snow profiles on the south aspect in the alpine, treeline, and below treeline elevation bands. The right column represents snow profiles at the same elevation bands, but on the north aspect. Colors represent grain type, which are denominated according to the international classification of Fierz et al. (2009)

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
