# Peer review of "Subgridding high-resolution numerical weather forecast in the Canadian Selkirk mountain range for local snow modeling in a remote sensing perspective"

_EGUsphere, 2023_

## Author Comment (AC1)

**Subgridding High Resolution Numerical Weather Forecast in the Canadian Selkirk range for local snow modelling in a remote sensing perspective**

REVIEWER 2

Billecocq et al. present a study introducing a subgridding method to downscale Numerical Weather Prediction model outputs to drive a detailed snowpack model, in order to produce spatially distributed estimates of SWE and snow microstructure properties at 100 m resolution. These estimates are necessary to provide a first guess of the snowpack structure to SWE retrieval algorithms in a perspective of remote sensing.

The paper covers a topic of interest for the cryospheric community and fits the scope of The Cryosphere. It is overall concise and reads easily, with helpful figures. It is an interesting contribution to that topic, but the following major issues need to be addressed.

Thank you for your valuable feedback and constructive comments. We have revised the manuscript accordingly.

The research plan remains somehow unclear in the paper. I would recommend to better define the research gap in the introduction, after a more complete literature review. What is the exact novelty of the paper? Almost no literature review is done on existing NWP downscaling methods and frameworks, while it seems to be identified as the main contribution of the paper. The definition of the research gap should then be followed by a clear exposition of the research questions that the study addresses. Discussion and conclusion could then be better structured to answer these research questions, based on the exposed methods and results. In the end, it could enable a better structure of the paper, enrich introduction and discussion sections and prevent reader's misunderstandings (which may be the cause of some of my comments).

The introduction structure has been restructured as suggested by the reviewer. It now features a literature review on NWP, and 3 defined research questions are formulated subsequently. The Discussion section has been significantly altered, with clearer statements of the achieved results, and a perspective on how these results will transfer to the field of remote sensing, which is the application domain for the proposed framework. The Conclusion section has been updated to feature answers to each research question, and it summarizes how the research presented in the paper contributes to answering each of them.

The validation of the snowpack simulations against AWS-driven snowpack simulations is problematic. First, they provide three point comparisons which cannot be representative of the domain. The lack of spatially distributed snowpack measurements and the absence of

more point measurements are understandable, but the comparison cannot really be considered as validation. Indeed, because of error compensations, a better weather input (from AWS measurements) does not necessarily provide a SWE or HS estimate by the model closer to actual snowpack observations. Additional snowpack measurements should be included for validation, and if not possible, the authors should be careful with the used words (e.g., "validation", "improve", "perform better", …). It also remains unclear if the HS measurements were used or not (i.e., are the reference station curves direct measurements or AWS driven simulations?). If so, it provides a first element of validation, but it should also be compared to the AWS driven simulations that are considered as references otherwise.

First, the wording has been updated according to the reviewer's comment everywhere in the manuscript.

Second, SR50 snow depth time series have been added as a proper validation tool for HS where available (Abbott and Fidelity). Moreover, the accuracy of the SNOWPACK model is now extensively discussed. It gives a good perspective on where and why AWS-SNOWPACK runs can be considered as validation, as well as a good view on where the biases of the model lie with respect to the reality. Previous work on that matter was conducted in our research lab, and we rely on the work of Madore et al 2018 and Madore et al 2022

https://www.tandfonline.com/doi/abs/10.1080/02723646.2018.1472984
https://www.frontiersin.org/articles/10.3389/feart.2022.898980/full

The spatial variability study is incomplete: six arbitrary points are not representative. Instead, pixels of the whole domain could be aggregated by topographic categories.

To complete the spatial variability, we added a new section to the results section domain-wide results aggregated by topographic categories. Furthermore, the intra-cell variability was assessed over the whole simulation, which is also presented in the results section for the 2018-2019 season. However, the model is crashing for season 2019-2020. The source of the bug is not trivial to identify, and we are still looking for a solution to this day.
Finally, the choice of these six arbitrary points is further justified, as they are located in the only cell of the domain that features an elevation range from below treeline to the alpine on both the north and south aspects. This makes this cell a perfect candidate to study the ability of the subgridding framework to provide meaningful spatial variability.

**Specific comments**

In general, the captions of all figures must be edited (in particular, from Figure 3 to the end). They need to be more descriptive of all represented variables.

Captions have been edited for every figure and table of the manuscript as recommended.

Abstract, l. 1-7: the part of the abstract about remote sensing could be shortened. It is only a context element, so it could be mentioned in one or two short sentences in the abstract.

A few sentences on the remote sensing background have been removed from the abstract, results have been updated, and an emphasis has been put on the perspectives and applications. Here is the updated version of the abstract.

Snow Water Equivalent (SWE) is a key variable in climate and hydrology studies. Yet, designing a SWE retrieval algorithm is not trivial, as multiple combinations of snow microstructure representations and SWE can yield the same radar signal. The community is converging towards forward modeling approaches using an educated first guess on the snowpack structure. However, snow highly varies in space and time, especially in mountain environments where the complex topography affects atmospheric and snowpack state variables in numerous ways. Automatic Weather Stations (AWS) are too sparse, and high-resolution Numerical Weather Predictions systems have a maximal resolution of 2.5 km × 2.5 km, which is too coarse to capture snow spatial variability in a complex topography. In this study, we designed a subgridding framework for the Canadian High Resolution Deterministic Prediction System. The native 2.5 km × 2.5 km resolution forecast was subgridded to a 100 m × 100 m resolution and used as the input for snow modeling over two winters in Glacier National Park, British Columbia, Canada. Air temperature, relative humidity, precipitation and wind speed were first parameterized regarding elevation using six Automatic Weather Stations. Alpine3D was then used to spatialize atmospheric parameters and radiation input accounting for terrain reflections and perform the snow simulations. Modeled snowpack state variables relevant for microwave remote sensing were evaluated against profiles generated with Automatic Weather Stations data and compared to raw HRDPS driven profiles. Overall, the subgridding framework improves on average the optical grain size (OGS) bias by 18\%, and the modelled SWE by 16\% with regards to simulations driven with raw HRDPS forecasts. This work could lead up to a 7 dB improvement in the snowpack SAR backscattering modelling, and hence provides the necessary basis for SWE retrieval algorithms using forward modeling in a Bayesian framework.

Abstract, l. 8-9 and Introduction l. 52-53: "too coarse to capture snow spatial variability in a complex topography". It always depends on the scale of variability you need to resolve, and so mostly depends on the modelling goal. "too coarse" should then be related to the application. The introduction needs to justify why this scale of 100 m is chosen, and why it is the appropriate resolution to resolve the spatial variability required by the remote sensing application.

No further justification was provided in the Abstract, but the Introduction has been modified as follows.

Moreover, HRDPS spatial resolution is too coarse to properly represent the spatial variability of atmospheric parameters and SWE in complex terrain. Indeed previous work on spatial variability of SWE and atmospheric parameters have shown that a scale break appears for SWE in the [50, 100] meters grid resolution interval, [100 m, 250 m] for wind exposure, [100 m, 180 m] for vegetation height, and [90 m, 100 m] for incoming solar radiations, leaving the optimal grid resolution at a 100 m for mountain processes \citep{grunewald2010, winstral2014, rittger2016a}. Finally, 100 m resolution ties in well with operational SAR satellites products and their processing pipelines, such as Sentinel-1 or TerraSAR-X, as well as future missions \citep{derksen2021}.

l. 45-46: detailed snowpack models could be a bit more extensively described. It could also be through a more complete description of the SNOWPACK model in the methods (e.g., when mentioning Alpine3D).

A more thorough description of the SNOWPACK/Alpine3D models has been added to the methodology section.

Alpine3D is a spatially distributed 3D model, which allows running the vertical 1D snow model SNOWPACK over an area, considering the spatial processes affecting atmospheric variables \citep{Lehning2006}. SNOWPACK is a detailed multi-layer thermodynamic finite-element model of snow microstructure and metamorphism. In this model, the snow microstructure is represented by four main variables: grain size, bond size, dendricity and sphericity for each snow layer. In addition, the model simulates several metrics of interest when monitoring the evolution of the snowpack, such as height of snow, SWE, density, optical grain size, or snow temperature \citep{Bartelt2002, Lehning2002}. To do so, the model is fed with three text files describing the weather parameters on the time domain of the simulation, the initial state of the soil layers on which the snow is going to develop (and initial snow layers if relevant), and finally the configuration of the simulation.

l. 48-50: also mention spatial representativity issues of AWS.

We agree with the reviewer, and this is what was meant behind the "local biases" in the original sentence. It has been rephrased in order to make this idea clearer to the reader:

However, they need human maintenance, are subject to outages, local biases, and usually undersample the spatial heterogeneity of the processes at stake, especially in complex terrain. As a result, AWS spatial interpolation in mountainous areas is not always accurate \citep{lundquist2019}.

l. 50-51: "high-resolution atmospheric models are known for their negative bias in precipitation". This statement is too generalized: is it the case for all high resolution atmospheric models, in all regions?

We agree with the reviewer that this statement is too generalized, it has been rephrased as:

The HRDPS model is known for its negative bias in precipitation.

l. 55-57: Why should this particular downscaling be discarded, regarding snow microstructure? It needs more justification.

These lines have been removed as part of the next comment answer. A proper literature review on NWP downscaling has been added.

l. 58-59: Many downscaling methods exist for driving snowpack models in complex terrain. The literature review about atmospheric downscaling could be largely extended (only one paper is cited). Secondly, the authors should justify clearly why existing downscaling methods are irrelevant for SWE and microstructure retrieval from snowpack models. The authors could be clearer about their research gap, to better identify the novelty compared to other recent studies. For example, Marsh et al. (2020) offer a modelling framework in Canada, including meteorological downscaling and the SNOWPACK model. Sharma et al. (2023) use dynamical downscaling of NWP with WRF within CRYOWRF (including the SNOWPACK model). A more complete literature review should enable the authors to better highlight the novelty of their study.

A paragraph has been added in the Introduction to present a literature review on NWP downscaling schemes. The niche where the proposed framework fits is hence better highlighted, and specific research questions have been added as a conclusion to the introduction to further underline the needs that are being addressed with this work.

[revised manuscript text omitted]

l. 79: Why not also including the melt season?
The global objective of this work is to provide a realistic first guess of the snowpack structure in the context of SAR remote sensing signal inversion algorithm development. At relevant frequencies (Ku-band, X-band, C-band), the snowpack becomes opaque to microwaves when wet. This is why the study focuses on the accumulation period.

l. 81 and l. 272: "round grains", "defragmented grains". Please stick to the official classification of grain shapes by Fierz et al. (2009). Here, respectively: "rounded grains", "decomposing and fragmented precipitation particles".

Corrections have been made directly in the manuscript.

Figure 2: I assume a typo ("VWS" for "VW")
Wind speed is referred to as VW everywhere in the SNOWPACK / MeteoIO /Alpine3D documentation. It stands for Velocity of Wind, as described in the official SMET format specification (*https://meteoio.slf.ch/doc-release/SMET_specifications.pdf*). This acronym was used everywhere in the manuscript out of homogeneity with the official specification.

Figure 2: please use the full word "microstructure"
The figure has been updated as requested.

l. 106: "Snow precipitation water equivalent" -> Snowfall
The wording was modified straight in the manuscript

l. 107-108: Were these precipitation boards located in areas free of wind-induced erosion or accumulation? This is worth mentioning.
Yes, they were placed in areas sheltered from the wind. It is now mentioned in the manuscript.

l. 112-122: This is all methods and results from Helbig and Löwe (2014), which should be cited. I would delete all these lines and equations and simply say Fsky is computed following Helbig and Löwe (2014).

l. 123-135: Once again, all of this is from Helbig et al. (2017), so not new. I don't see the need to reproduce all equations here if the authors simply say they use their downscaling method.

The equations were added for the reader to have every equation used in the subgridding paper in the same document to ease the reproduction of the methodology. However, we understand that it is not necessary to reproduce them here. The equations have been removed, and Helbig and Löwe (2014) is now cited. Changes were made directly in the manuscript.

l. 141: "ILWR was spatialized using IDW". ILWR is strongly dependent on terrain elevation (e.g., Marty et al. (2002) found a climatological vertical gradient of - 29 W/m²/km in the Alps). I would assume a simple IDW is not sufficient to downscale ILWR in complex terrain, or am I missing something?

This point has been addressed by spatializing ILWR using the IDW_lapse algorithm. We used the same lapse-rate as reported by Marty et al. because there are not enough ILWR measuring stations in our study area to compute our own local gradient. The Method, Results, and corresponding figures have all been updated accordingly.

l. 141-142: "All the algorithms mentioned above are a part of the MeteoIO library, which is integrated into the Alpine3D model". It is a bit unclear here what is novel in this study ("We designed a logarithmic regression (…)", l. 98) and what is already existing in Alpine3D. Clarification is necessary.

With the added paragraphs in the introduction we believe that the different points of novelties introduced in this paper should now be clearer to the reader. The sentence in l.141 has been rephrased to:

All the spatial interpolation algorithms mentioned above are a part of the MeteoIO library, which is integrated into the Alpine3D model.

This makes the point clearer that novelty with regards to NWP subgridding lies in the parameterization of the HRDPS into the Virtual Station Array. Second point of novelty lies in the thorough evaluation of the snowpack parameters on the point scale and on the whole simulation domain, which had never been performed before for such a simulation framework.

l. 143: The snowdrift scheme is probably turned off since wind-driven redistribution is parameterized by a precipitation multiplier? It could be worth mentioning.

Yes, the snowdrift is turned off. This is now mentioned in the text.

l. 144: "considering the spatial processes affecting atmospheric variables". Do you mean the atmospheric downscaling already described above? Please reformulate or clarify.

Here we only meant that Alpine3D has some built-in features to consider the effect of topography on the input array of atmospheric variables.
The sentence has been rephrased as follows:

Alpine3D is a spatially distributed 3D model, which allows running the vertical 1D snow model SNOWPACK over an area while taking into account the spatial processes affecting the input atmospheric variables, such as terrain shadowing (Lehning et al., 2006).

l. 147-150: Why choosing individual points? How have they been chosen? They are not necessarily representative. Why not aggregating values by categorical topographic areas instead? What's the model slope at the chosen points? It can have a significant impact on ISWR differences between North and South.

A few sentences have been added to justify the choice of choosing these specific individual points. A Table now presents topographic characteristics for each point as well.

To assess the spatial variability capacity of the subgridding framework, the model was ran on the whole simulation domain, and we also generated outputs at six points within the same cell for intra-cell variability assessment. The specific cell was chosen because it is the only cell in the simulation domain that features a north and south slope with elevations ranging from below treeline to the alpine on both aspects. No glacier is present in the area. Table \ref{tab:spatial_variability_points} summarizes the topographic characteristics for the chosen intra-cell spatial variability points.

Furthermore, as stated earlier, the spatial variability analysis has now been enriched with plots showing snow parameters over the whole domain aggregated by topographic categories.

l. 152-153 and further: as mentioned in the general comments, I would not call a comparison of SGF-SNOWPACK vs AWS-SNOWPACK a validation of snowpack simulations. Some studies have shown that a "better" weather input could degrade the metrics of snowpack simulations, because of error compensations in the interplay of weather input and modelled snowpack processes. Please be careful with the wording "validation", "better", etc., to qualify the comparison. "Simulation A is closer to Simulation ref than Simulation B" would be more accurate.
The wording has been corrected everywhere in the manuscript. A paragraph has been

added to the Discussion section where the comparison/validation of the subgridding framework with AWS driven SNOWPACK runs is discussed and justified.

l. 186: as mentioned before, simply say "HRDPS tends to"...
The modification has been done in the manuscript.

l. 192: "a mean layer-by-layer bias for density and OGS". This needs to be clarified a bit. Is it a mean value for a snowpack profile where a 1 mm layer would weight the same as a 50 cm layer? Is there a weighting?

The mean layer-by-layer bias is computed between the warped (i.e. aligned) profile and the reference profile (c.f. 3.3.). Being "aligned" on the reference, the warped profile has now the same HS as the reference (which is the ground truth). Moreover, at this point in the framework, the two profiles are gridded on the same elemental grid described in 3.3.. As a result, at this stage "layers" are just vertical chunks of snow with the same thickness which have no link to the physical layering of the snowpack. Therefore, the layer-by-layer mean bias is actually equivalent to the bulk density bias. The wording has been changed to "mean bias" over the whole manuscript to avoid confusion.

l. 193: "Height of Snow and SWE were visually assessed". Why not metrics?
Height of Snow and SWE were assessed using the Nash-Sutcliffe model efficiency coefficient \citep{NashStucliffe1970} over the two seasons. SWE was assessed against the AWS driven SNOWPACK runs, and HS was evaluated against SR50 HS measurements at Abbott and Fidelity station (unfortunately, there is no HS measuring device at Hermit station).

l. 195-197: The calculation of NSE may not need to be described here.

The equation was removed from the manuscript.

l. 220-222: The bias should be computed as model - reference, so that a positive bias would mean an overestimation, and it should be the case for all variables. It would avoid unnecessary confusion.
Bias computation has been corrected according to this comment, and corresponding figures and interpretations have been updated.

Figure 3: The vertical labeling is somewhat confusing. Perhaps simply write TA bias (°C), RH MAE (%), etc.?
The vertical labelling has been updated as suggested.

l. 225-226: It could be worth mentioning it corresponds to very shallow snowpacks.

The season begins with average similarity values (around 0.5), then it plummets to low values in mid-October (<0.5) when the snowpack is non-existent or still very shallow. By mid-November, the similarity then improves to higher levels of similarity, and stays relatively constant for the rest of the season (0.6 <sim <0.8).

Figure 4, green curve: I would not call it Station, since it could be confusing for the reader assuming it's a station observation. More understandable labels could be, for example: SGF-SNOWPACK, HRDPS-SNOWPACK, AWS-SNOWPACK. Or is it actually the HS measurement in green? If so, the AWS driven SNOWPACK simulation should also be represented since it is the reference for the other metrics. Moreover, in Figure 6, the green curve is called SWE Station, even though there is no SWE measurement at the station (according to Table 1). This is very confusing and should be clarified.
Labels have been modified as suggested.

l. 239-241: Isn't it rather related to the fact that the metrics are computed over very shallow snowpacks?

Yes, it is definitely the case. This is discussed in the Discussion section as we feel it is more of an interpretation than an observation. We slightly modified the sentence to prevent any confusion when reading the sentence.

Considering the strong fluctuations of the similarity signal in the fall for both seasons, the first three months (September to November included) were considered as a spin-up phase for the model to initiate a proper snowpack.

l. 241-242: "the numerical analysis of the results was carried out starting on the first of December." Is it also the case for the similarity metrics exposed a few sentences earlier? If so, please clarify.

Yes, it is. This sentence is now appearing before the similarity metrics in order to clarify.

l. 242: "HRDPS and the subgridding framework are slightly overestimating OGS". The concision of this paper is overall appreciated, but it might be clearer to use formulations such as HRDPS-SNOWPACK and SGF-SNOWPACK, because snowpack-related variables are not an output of HRDPS and the subgridding framework.
We agree with the reviewer and references to snow simulations have been updated throughout the manuscript.

l. 246-253: As far as I understand, the mean density bias is a mean of the biases of all layer densities, i.e., not equivalent to the bulk density bias. It would deserve to be better clarified. It would also be necessary to justify why this layer mean is chosen over a bulk density. It seems very dependent on the layering?

The mean layer-by-layer bias is computed between the warped (i.e. aligned) profile and the reference profile (c.f. 3.3.). Being "aligned" on the reference, the warped profile has now the same HS as the reference (which is the ground truth). At this point in the framework, the two profiles are gridded on the same elemental grid described in 3.3.. As a result, at this stage "layers" are just vertical chunks of snow with the same thickness which have no link to the physical layering of the snowpack.  Therefore, the layer-by-layer mean bias is actually equivalent to the bulk density bias. The wording has been changed to "mean bias" over the whole manuscript to avoid confusion.

l. 255-256: "As a result (…)". This logical link is unclear. The SGF could also reduce the PSUM bias. Please reformulate or clarify.

The paragraph was reformulated, it includes the new numbers from the new simulations, comparisons with the snow height measured by the SR50 at Abbott and Fidelity, which clarifies the link between the SGF, PSUM bias, and HS errors.

Finally, SGF-SNOWPACK mean error on modeled HS is 35 cm in 2018-2019 (20 cm improvement when compared to HRDPS-SNOWPACK), and 29 cm in 2019–2020 (29 cm improvement) when compared with the SR-50 measurement at Fidelity. For reference, the modelled HS with AWS-SNOWPACK shows a mean error of 8 cm in 2018-2019 and 14 cm in 2019-2020. However, SGF-SNOWPACK seem to degrade the quality of the HS modelling with regards to HRDPS-SNOWPACK when compared to the SR50 at Abbot station, overestimating HS for each season. Yet, the Station-SNOWPACK run at Abbott shows a high discrepancy with the SR-50 measurements as well, overestimating HS as well (especially in 2018-2019). Finally, HS remained relatively unchanged at Hermit for 2018–2019 as the framework did not bring a substantial improvement when compared to the Station-SNOWPACK modelled HS.

l. 263: The observed altitudinal temperature gradient is the reflect of the lapse rate chosen for TA downscaling (l. 137). There is no proof here it is realistic.

This line has been rephrased as :

First, the lapse-rate applied for TA downscaling and spatialization respects the general rule of thumb that TA should get colder with elevation.

l. 264-265: "slightly lower temperatures in the north aspects". Figure 7 shows the contrary for TA in Alpine area (warmer TA on North slopes vs South slopes, consistently throughout the season). Any explanation? Or is it a plotting error?

In the selected spatial validation cells, the South slope alpine pixel has an elevation of 2197 m whereas the North one has an elevation of 2079 m. With a hundred meters gap in elevation and considering the TA lapse-rate correction, it makes sense that the South aspect shows consistently lower temperatures. This is now clear to the reader thanks to the included Table in the methodology section, and it is now reflected in the text and justified to the reader:

However, the selected point on the North aspect is a 100 m lower than the South aspect point, and as a result, the figure shows slightly warmer temperatures consistently throughout the season on the North aspect.

l. 268-269: "the altitudinal precipitation rate gradient is also respected by the subgridding framework, with precipitation rates getting higher with elevation". Is there any incremental improvement compared to the original HRDPS gradient?

The original HRDPS cells cover an area of 2.5 km x 2.5 km. Each 100 m x 100 m DEM cell underneath would have received the same amount of snowfall, without any altitudinal rate modifier despite a strong elevation gradient within the same HRDPS cell. Moreover, the HRDPS precipitation gradient is only relevant at the scale of the surface model used to run the atmospheric model, I do not think that a comparison is relevant in this case, as we are evaluating the ability to subgrid the weather parameters within one HRDPS cell in a physically sound way, and perform spatialized snow simulation out of these newly generated weather inputs.

l. 270: What is the reason for simulating the snowpack in forested areas (in a remote sensing perspective), if the forest snow processes, which have a strong impact on the snowpack, are not represented?

We agree with the reviewer that there is limited interest in simulating the snowpack in forested areas in a remote sensing perspective. The difficulty of accurately modelling both the snowpack and radiative transfer under trees and snow makes for a particularly challenging problem. However, we have chosen to tackle the entire elevation range within our study area out of completeness, in order to assess how the subgridding framework is performing on the entire domain of the simulation.

l. 271-272: "The wind erosion effect on the snowpack is also well represented, as dominant winds are blowing from the South / South-West. As a result, the south aspect profiles show more defragmented grains (dark green) on the surface". I am not sure I understand this cause-consequence. As far as I understood, wind-induced snow transport is represented by a precipitation multiplier. Consequently, associated effects of snowdrift on snow microstructure are not represented. Or am I missing something? Please clarify.

The word "erosion" here has not been used appropriately by the authors and is certainly the cause of the misunderstanding. Lines 271 - 274 in the original manuscript have been modified as such:

 The wind effect on the snowpack is also well represented in the simulations. Indeed, dominant winds are blowing from the South / South-West, and as a result southern slopes are affected by stronger winds (fig. 7). In the SNOWPACK model, grain type is a function of dendricity and sphericity, two parameters governed by the temperature gradient within the snowpack. As the surface temperature is altered by surface winds, precipitation particles (lime green) on the south aspects tend to metamorphose faster into decomposing and fragmented precipitation particles (dark green) than in the northern aspects, especially in the alpine.

l. 274: "slower settlement". This needs to be proven, it is not obvious when looking at the figure.

This sentence has been removed (c.f. previous comment)

l. 274-275: An extension of the simulations to Spring would be interesting to assess the melting processes and differences between North and South slopes.

Because of the SWE remote sensing orientation of this work, we focused on the dry snow season. Even though we agree with the reviewer that it would be interesting, we argue that it falls out of the scope of the present paper.

l. 280: "a considerable improvement". With respect to the results exposed in Figure 3, this assertion could be more nuanced.
The sentence has been rephrased to "a noteworthy improvement", which seems more appropriate indeed.

Figure 8: Please provide the label for grain type color codes.
A colorbar has been added to Figure 8 and 10 with grain type color codes.

l. 290-295: It could be clarified.
These sentences have been modified as follows:

However, the HRDPS TA presents a negative bias at these 3 stations (i.e., HRDPS is colder than station measurements), and all 3 stations are higher than the nominal elevation of their overlying HRDPS cell. As a result, a naive Inverse Distance Weighting lapse-rate spatialization scheme would introduce even more bias in the system, aggravating the

original error. The introduction of our logarithmic bias correction in the parameterization step of the framework reduces the bias in TA before the IDW spatialization. Thus, the logarithmic lapse-rate parameterization allows reducing the error in the subgridding scheme and keeps it within a physically meaningful interval regarding spatial resolution.

l. 300: The discussion could explore further the reasons why and how simulated microstructure parameters are modified, with the modified input.

The new version of the discussion gives more details on processes that affect snow parameters. To some extent, it is also discussed in section 3.3. on the evaluation of snow simulations:

As precipitation is usually underestimated by HRDPS, HS should be underestimated as well, which should impact the overburden pressure on basal layers. This might result in a small negative bias on density with regards to AWS driven SNOWPACK runs, depending on the amount of missing snow. For OGS, the temperature gradient in this region is low and metamorphism mainly happens through gravitational settling, leading to little variability in OGS in the snowpack \citep{madore2018}. As a result, we do not expect much impact of the inflation approach on this microstructure parameter, as the main discrepancies should come from offsets in rain-on-snow modeling, and melt/percolation events.

l. 306-310: The authors could provide typical snow OGS values to give an idea of the relative change.

A comparison with typical values of OGS in the area has been added :

Furthermore, the subgridding framework allows decreasing by 0.04 mm the OGS overestimation compared with raw HRDPS simulations (averaged over all sites and seasons). Optical diameter usually ranges from 0.1 mm to 0.4 mm in this region, the subgridding framework can improve the modelled OGS by 10 to 40 percent on average.

l. 311: "wind transport in the alpine is likely exaggerated". This statement needs to be justified.

To account for the other comments, the discussion has been substantially modified, and this phrase no longer appears in it.

l. 314-323: In the current state, this paragraph is more a perspective for the conclusion section.

We agree to this comment, the paragraph has been moved to the conclusion section.

l. 325-326: See previous comments about clearer identification of the research gap.

To better underline where the contribution and novelty of the paper, this phrase has been modified as follow:

The work presented in this study aims at solving the present need in the snow remote sensing community for both the design and the evaluation of an atmospheric model subgridding framework to perform snow modelling in the context of coupling with a SAR signal inversion routine.

l. 341: The word "stabilize" may be misused, since the SNOWPACK simulations are probably not strictly speaking "unstable"?

The word "simulation" is actually a typo here. "Similarity" was meant instead and the error went through our correction process before submission. This sentence has been rephrased as follows:

In this context, the first three months (September to November) of snow simulations should be considered as a spin-up phase for the snow model, as discrepancies between reality and simulations are critical before the snowpack is properly established and the similarity stabilizes in December and onward.

---

## Author Comment (AC2)

**Subgridding High Resolution Numerical Weather Forecast in the Canadian Selkirk range for local snow modelling in a remote sensing perspective**

**REVIEWER 1**

Billecocq at al present an interesting study on refining the spatial resolution of meteorological forcings to feed a detailed snow model (ALPINE3D). The ultimate goal is to test whether this refinement improves the representation of snow microstructure as relevant to SWE retrieval algorithms from satellite. Results show improvements for the optical grain size and SWE for two seasons of data in the Glacier National Park in Canada.

Overall, the study is well conceived and the paper is well written and concise. The topic is relevant and within the scope of TC. At the same time, there are in my opinion a number of major and minor points that should be addressed before publication. Thus I am recommending a major revision.

*Thank you for your valuable feedback and constructive comments. We have revised the manuscript accordingly.*

The first major comment is that all snow evaluations are performed using simulated (not observed) time-series at only three locations over the study area. In the discussion, authors are clear on this being a limitation of their study (lines 312). However, I think this point should be better addressed throughout the manuscript as the main critical aspect of this work. Ideally, the best solution would be to include observations in this evaluation exercise, but it may be that such observations are not available at the considered study site. So I see two potential alternatives: (1) include results from other regions where such data are available, and/or (2) better discuss accuracy and precision of SNOWPACK simulations forced using AWS data using reference literature (e.g., https://tc.copernicus.org/articles/9/2271/2015/)

*This comment was addressed in several ways. The Abbott and Fidelity stations are equipped with SR50 instruments to measure snow depths. Time series were included and the SNOWPACK simulations were evaluated against it. Moreover, a paragraph was added in the discussion to discuss the accuracy and precisions of the SNOWPACK simulations. Previous work on that matter was conducted in our research lab, and we rely on the work of Madore et al 2018 and Madore et al 2022*

*https://www.tandfonline.com/doi/abs/10.1080/02723646.2018.1472984*
*https://www.frontiersin.org/articles/10.3389/feart.2022.898980/full*

Second, results are promising with regard to snow depth / SWE, but quite incremental when looking at the optical grain size and density (see line 16 and then the results section). The

same could be said with regard to weather forcing data, where a clear benefit of downscaling is evident (in my opinion) for radiation and humidity, while results for temperature and precipitation are mixed. While authors are again clear on this (see the discussion section for example), and while I totally see the main point of novelty provided by the authors (line 325), I am still wondering what is the significance of this work for the global audience of TC given these mixed results and the fact that authors focused on a comparatively small region and two years of data. To overcome this, I am proposing to (1) include a clearer justification regarding the choice of this study region, including why it is important for the global readership of TC; **(2) significantly expand the Discussion section with much clearer statements of the main findings, implications, and future steps in view, and in the context, of the relevant literature**; (3) ideally, include specific research questions in the Introduction to further generalize findings.

The Study Site section has been enriched with a few sentences justification on why this study site is relevant for such a study.

This study was conducted in the Rogers Pass area of Glacier National Park (GNP), British Columbia, Canada (Figure \ref{fig:GNP_map}), which is part of the Selkirk range in the Columbia Mountains. The pass is used as a transportation corridor by the Trans Canada Highway and the Pacific Railway, making it the busiest transport corridor in Western Canada \citep{bellaire2016}. The pass is exposed to 144 avalanche paths, and as a result, Rogers Pass hosts the largest avalanche control operation in Canada \citep{delparte2008a}. The operation has been ongoing since 1965 and the site has been used as a snow research site ever since, making this area the longest record of mountain snow in Western Canada \citep{fitzharris1987, bellaire2016, madore2022}.

The Discussion section has been significantly altered, with clearer statements of the achieved results, and a perspective on how these results will transfer to the field of remote sensing, which is the application domain for the proposed framework.

Finally the Introduction has been modified to further outline the need for the proposed research and where it stands with regards to the state-of-the-art. 3 Research questions emerge from the paragraph and then answered in the Discussion and Conclusion sections.

1. How do subgridded HRDPS forecasts compare to reference Automatic Weather Stations in the simulation domain ?
2. Do the resulting atmospheric forcings lead to an improvement in snowpack modelling, especially for critical snow parameters in remote sensing applications ?
3. Which degree of spatial variability with regards to snow parameters can be reached by such a subgridding framework ?

**Minor / specific comments**

- Abstract: in my view, the abstract focuses too extensively on background information (up to line 10). I would recommend summarizing this background information to focus on the main findings and implications

A few sentences on the remote sensing background have been removed from the abstract, results have been updated, and an emphasis has been put on the perspectives and applications.. Here is the updated version of the abstract.

Snow Water Equivalent (SWE) is a key variable in climate and hydrology studies. Yet, designing a SWE retrieval algorithm is not trivial, as multiple combinations of snow microstructure representations and SWE can yield the same radar signal. The community is converging towards forward modeling approaches using an educated first guess on the snowpack structure. However, snow highly varies in space and time, especially in mountain environments where the complex topography affects atmospheric and snowpack state variables in numerous ways. Automatic Weather Stations (AWS) are too sparse, and high-resolution Numerical Weather Predictions systems have a maximal resolution of 2.5 km × 2.5 km, which is too coarse to capture snow spatial variability in a complex topography. In this study, we designed a subgridding framework for the Canadian High Resolution Deterministic Prediction System. The native 2.5 km × 2.5 km resolution forecast was subgridded to a 100 m × 100 m resolution and used as the input for snow modeling over two winters in Glacier National Park, British Columbia, Canada. Air temperature, relative humidity, precipitation and wind speed were first parameterized regarding elevation using six Automatic Weather Stations. Alpine3D was then used to spatialize atmospheric parameters and radiation input accounting for terrain reflections and perform the snow simulations. Modeled snowpack state variables relevant for microwave remote sensing were evaluated against profiles generated with Automatic Weather Stations data and compared to raw HRDPS driven profiles. Overall, the subgridding framework improves on average the optical grain size (OGS) bias by 18\%, and the modelled SWE by 16\% with regards to simulations driven with raw HRDPS forecasts. This work could lead up to a 7 dB improvement in the snowpack SAR backscattering modelling, and hence provides the necessary basis for SWE retrieval algorithms using forward modeling in a Bayesian framework.

- line 8: this maximal resolution of 2.5 km is likely specific for Canada datasets (?)
Yes, this sentence has been rephrased as : "Moreover, HRDPS spatial resolution is too coarse to properly represent [...]"

- line 31: this statement on models yielding biased estimates of SWE at high elevation is likely too generic. Several correction approaches in this regard have been documented, but results are very site specific (which is in my opinion the actual main challenge here)
This statement has been rephrased as:

Moreover, both observations from passive microwaves and modeling efforts yield negative biases when estimating mountain or deep-snow SWE on the global scale \citep{vuyovich2014, wrzesien2018, pulliainen2020}.

- line 49: I think that the main reason why AWS spatial interpolation in complex terrain is not accurate is because AWS systems undersample the real spatial heterogeneity of the processes (which is not mentioned here)
We agree with the reviewer, and this is what was meant behind the "local biases" in the original sentence. It has been rephrased in order to make this idea clearer to the reader:

However, they need human maintenance, are subject to outages, local biases, and in most cases undersample the spatial heterogeneity of the processes at stake (one would need an exceptionally dense array of weather stations), especially in complex terrain. As a result, AWS spatial interpolation in mountainous areas is not always accurate \citep{lundquist2019}.

- line 51: AWS systems are also prone to undercatch and so underestimation of precipitation (this is one of the main reasons why I think it would be ideal to include actual measurements of snow properties in the evaluation).
We agree with the reviewer and SR50 HS measurements have been added to the data presented in the paper. The sentence has not been modified here, as the idea behind it is to show the reader that AWS have biases (idea developed in the previous sentence), but so do atmospheric models.

- line 76: please avoid reporting units in italics
This has been corrected everywhere in the manuscript.

- Figure 1: consider including a DEM here
The figure has been modified to include elevation information for the reader.

- Table 1 and all other captions: please consider defining acronyms in captions for diagonal readers
The Table's caption has been edited.

- line 88: please specify "most of Canada"
This is how Environment and Climate Change Canada describes the HRDPS forecast. It covers the vast majority of Canada, only leaving out the most northern islands or the Arctic archipelago.

- line 100 to 110: correction factors for temperature, radiation, and precipitation are very succinctly presented, to the extent that repeatability of these experiments may be difficult. How was Eq. 1 derived (what data were used? What period? What optimization approach?). Same for Eq. 2. Why was Equation 1 used for dew point temperature too?
Descriptions for the TA and PSUM correction equations have been updated with more

details. For Relative Humidity, no detail was added as the methodology we used is the exact same as the one presented in the cited reference (Liston and Elder, 2006). They present the conversion of RH to dew point temperature which is then corrected according to an elevation lapse-rate equation to account for elevation discrepancy. Then dew point temperature is converted back to RH. Having developed a correction equation for temperature more suited to our study site, in our opinion iit makes only sense to use this one rather than the standard lapse-rate equation.

Updated description for the TA correction equation:
Using all weather stations in the Park, bias in air temperature was found to have a non-linear relationship with the elevation difference between the station elevation and the original HRDPS cell elevation over the 2018-2020 period. A training set was generated by randomly selecting 75\% of this dataset uniformly across elevations, and the remaining 25\% served as validation set. The data was transposed into logarithmic space to perform a linear regression. The resulting logarithmic fit was then applied over the TA dataset when the elevation difference between the Virtual Weather Station and its overlying HRDPS cell was over 100 m.

Updated description for the PSUM equation:
Snowfall was first parameterized using an elevation lapse-rate correction. This lapse rate was computed by performing a simple linear regression of precipitation as a function of elevation. We used a dataset of four weeks of manual SWE measurements on four conventional HN24 precipitation boards placed between 1330 m and 1920 m at Mt Fidelity, all placed in areas sheltered from the wind.

- Line 136: why did you first use a 20-m DEM and now a 100-m one?
The 20 m DEM is used for the parameterization of the Virtual Station array. The 100 m DEM is the grid upon which the spatialized snow simulations will be computed. It would not make sense to run spatialized snow simulations at a 20 m resolution both from a computation performance stand-point and a spatial variability process. A paragraph has been added in the introduction to further justify this resolution choice for snow simulations:

Moreover, HRDPS spatial resolution is too coarse to properly represent the spatial variability of atmospheric parameters and SWE in complex terrain. Indeed previous work on spatial variability of SWE and atmospheric parameters have shown that a scale break appears for SWE in the [50, 100] meters grid resolution interval, [100 m, 250 m] for wind exposure, [100 m, 180 m] for vegetation height, and [90 m, 100 m] for incoming solar radiations, leaving the optimal grid resolution at a 100 m for mountain processes \citep(grunewald2010, Winstral2014, Rittger2016). Finally, 100 m resolution ties in well with operational SAR satellites products and their processing pipelines, such as Sentinel-1 or TerraSAR-X, as well as  future missions \citep(Derksen2021).

- Section 3.3: the inflation approach is clear, and I generally agree with this. At the same time, microstructural parameters are (to some extent) dependent on SWE and HS (via overburden pressure and temperature gradients, for example). If authors agree with this, I would add some discussion on how this could impact these results.

We agree with the reviewer and these concerns are now raised and discussed at the end of this section :

As precipitation is usually underestimated by HRDPS, HS should be underestimated as well, which should impact the overburden pressure on basal layers. This might result in a small negative bias on density with regards to AWS driven SNOWPACK runs, depending on the amount of missing snow. For OGS, the temperature gradient in this region is low and metamorphism mainly happens through gravitational settling, leading to little variability in OGS in the snowpack \citep{madore2018}. As a result, we do not expect much impact of the inflation approach on this microstructure parameter, as the main discrepancies should come from offsets in rain-on-snow modeling, and melt/percolation events.

---

## Referee Report (RR1)

The authors provided significant improvements to the manuscript by addressing most of the comments. The manuscript is also better introduced and contextualized. I still have a few minor comments and some technical corrections, described hereafter. It could be useful to carefully revise the English writing or make it proofread it by a native speaker.

**Comments about the point-to-point response**

(Citations of the response in red, my answers in black)

"l. 79: Why not also including the melt season?

The global objective of this work is to provide a realistic first guess of the snowpack structure in the context of SAR remote sensing signal inversion algorithm development. At relevant frequencies (Ku-band, X-band, C-band), the snowpack becomes opaque to microwaves when wet. This is why the study focuses on the accumulation period."

It would be worth mentioning in the text.

"Figure 2: I assume a typo ("VWS" for "VW")

Wind speed is referred to as VW everywhere in the SNOWPACK / MeteoIO /Alpine3D documentation. It stands for Velocity of Wind, as described in the official SMET format specification (https://meteoio.slf.ch/doc-release/SMET_specifications.pdf). This acronym was used everywhere in the manuscript out of homogeneity with the official specification."

The VWS acronym has not been corrected to VW ("VWS parametrized").

"l. 263: The observed altitudinal temperature gradient is the reflect of the lapse rate chosen for TA downscaling (l. 137). There is no proof here it is realistic.

This line has been rephrased as : First, the lapse-rate applied for TA downscaling and spatialization respects the general rule of thumb that TA should get colder with elevation."

The new formulation is also not very satisfactory. The fact that temperature decreases with elevation is not a result, but simply the direct consequence of the chosen lapse-rate.

"l. 270: What is the reason for simulating the snowpack in forested areas (in a remote sensing perspective), if the forest snow processes, which have a strong impact on the snowpack, are not represented?

We agree with the reviewer that there is limited interest in simulating the snowpack in forested areas in a remote sensing perspective. The difficulty of accurately modelling both the snowpack and radiative transfer under trees and snow makes for a particularly challenging problem. However, we have chosen to tackle the entire elevation range within our study area out of completeness, in order to assess how the subgridding framework is performing on the entire domain of the simulation."

This point remains quite unclear to me. As far as I understand, authors consider somehow "virtual open terrain" below treeline to cover a full elevation range. It should be more explicitly stated in the manuscript, together with motivations for doing so.

"l. 271-272: "The wind erosion effect on the snowpack is also well represented, as dominant winds are blowing from the South / South-West. As a result, the south aspect profiles show more defragmented grains (dark green) on the surface". I am not sure I understand this cause-consequence. As far as I understood, wind-induced snow transport is represented by a precipitation multiplier. Consequently, associated effects of snowdrift on snow microstructure are not represented. Or am I missing something? Please clarify.

The word "erosion" here has not been used appropriately by the authors and is certainly the cause of the misunderstanding. Lines 271 - 274 in the original manuscript have been modified as such: The wind effect on the snowpack is also well represented in the simulations. Indeed, dominant winds are blowing from the South / South-West, and as a result southern slopes are affected by stronger winds (fig. 7). In the SNOWPACK model, grain type is a function of dendricity and sphericity, two parameters governed by the temperature gradient within the snowpack. As the surface temperature is altered by surface winds, precipitation particles (lime green) on the south aspects tend to metamorphose faster into decomposing and fragmented precipitation particles (dark green) than in the northern aspects, especially in the alpine."

The grain type could be affected by many other parameters. In the present state, this conclusion is not sufficiently backed.

**Comments about the tracked change manuscript**

(Line numbers referring to the tracked change manuscript)

**Abstract**

l.3: "SWE retrieval". After deletion of the two previous sentences, we don't know anymore we're talking about satellites. Just mention quickly it's retrieval algorithm for remote sensing.

l. 7-9: "Automatic Weather Stations (AWS) are too sparse, and high-resolution Numerical Weather Predictions systems have a maximal resolution of 2.5 km × 2.5 km, which is too coarse to capture snow spatial variability in a complex topography." This statement is only applying to your study area which is not mentioned at this point.

l. 14-16: "profiles generated with Automatic Weather Stations data" is a bit unclear, please be more specific if it's simulated profiles driven by AWS data, or measurements. Similarly, rather say simulated profiles driven by raw HRDPS data.

l.19: SAR is not defined. On the contrary, you don't need to add the acronyms "(AWS)" and "(OGS)" there if you don't reuse them in the abstract.

**1. Introduction**

l. 54: you mention the HRDPS model but we don't know yet you're working on Canada. You could mention for example: "HRDPS, performing numerical forecasts at 2.5 km resolution over Canada (…)" or something like that.

l.57: the acronym NWP has not been explained at this point.

l. 57-94: The addition of a more complete literature review in the introduction is appreciated. However, the writing of this new paragraph could sometimes be clearer and more concise. There are many details which are not so necessary about some studies in the literature review (e.g. the part about the CHM model). The literature review should highlight the specificities of these studies regarding the focus of the present study and potential differences of methods. For example, how CHM snowpack simulations were compared to LIDAR data is not relevant in the context of a paragraph focusing on the atmospheric downscaling.

L. 80: missing reference.

**2. Study area**

l. 126: I count 8 AWS in the Park on Figure 1. Also please mention you only use six of them in your study area, as in Table 1.

**3. The Numerical Weather Predictions downscaling processing chain design**

l.148 : for clarity, perhaps you can add: "These parametrizations are described hereafter."

l. 167: rather say you downscale ILWR through a correction using the lapse rate highlighted by Marty et al. (2002). They did not literally introduce a correction.

l.195, and every other occurrence: precipitation (no s)

l. 200: repetition of "affecting"

l. 211: "whole"

l. 254: "tends"

**4. Results**

l.348 and 350: "AWS-SNOWPACK" should be used instead of "Station-SNOWPACK" I assume?

Figure 10 is not so easy to read: in particular, what are the x axis legends? Pixels "names"? Maybe the authors could think of a more "reader-friendly" figure conveying the same message.

**5. Discussion**

l.433: "whole"

l.433: "idealized" is probably not the best choice of word for a real site.

l. 431-435: more specifically, what parameterization of SNOWPACK are you talking about, and how could it vary with elevation?

l. 444: "HRDPS-SNOWPACK simulations"

l. 444-445: repetition

l. 454-455: unit error? The SWE bias is more likely in mm than cm.

l. 465 and 467: once again, is it really cm when talking about SWE (usually expressed in mm or kg/m$^2$)?

**6. Conclusion**

I would not repeat the research questions, but simply their answers with 1/2/3.

l. 512: "atmospheric variables"

---

## Author Response (AR2)

**Subgridding High Resolution Numerical Weather Forecast in the Canadian Selkirk range for local snow modelling in a remote sensing perspective**

First, we would like to thank the reviewer for his or her valuable feedback and comments which greatly helped improve the manuscript. We have revised the document according to the reviewer's minor comments. The present document summarizes our answer to the Referee report document. The original comments are in red, the reviewer's comments are in green. and our final answer is in blue. Moreover, the english was proofread after the science and technical corrections were made to the manuscript.

**Comments about the point-to-point response**

"l. 79: Why not also including the melt season?
The global objective of this work is to provide a realistic first guess of the snowpack structure in the
context of SAR remote sensing signal inversion algorithm development. At relevant frequencies (Ku-
band, X-band, C-band), the snowpack becomes opaque to microwaves when wet. This is why the
study focuses on the accumulation period."
It would be worth mentioning in the text.

This information has been added to the manuscript (now l. 112 in the revised manuscript)

"Figure 2: I assume a typo ("VWS" for "VW")
Wind speed is referred to as VW everywhere in the SNOWPACK / MeteoIO /Alpine3D documentation.
It stands for Velocity of Wind, as described in the official SMET format specification (https://meteoio.slf.ch/doc-release/SMET_specifications.pdf). This acronym was used everywhere in the manuscript out of homogeneity with the official specification."
The VWS acronym has not been corrected to VW ("VWS parametrized").

The VWS acronym the reviewer is referring to in Fig 2 stands for "Virtual Weather Stations", this is why we missed it in the first round of revision. This is obviously unclear to the reader, so the figure has now been modified and the acronym has been removed. We apologize for the misunderstanding.

"l. 263: The observed altitudinal temperature gradient is the reflect of the lapse rate chosen for TA
downscaling (l. 137). There is no proof here it is realistic.
This line has been rephrased as : First, the lapse-rate applied for TA downscaling and spatialization respects the general rule of thumb that TA should get colder with elevation."

The new formulation is also not very satisfactory. The fact that temperature decreases with elevation
is not a result, but simply the direct consequence of the chosen lapse-rate.

This sentence has been rephrased to :
First, as a direct consequence of the applied lapse-rate for TA downscaling and spatialization, the general rule of thumb that TA should get colder with elevation is respected.
(l. 300 in the revised manuscript)

"l. 270: What is the reason for simulating the snowpack in forested areas (in a remote sensing perspective), if the forest snow processes, which have a strong impact on the snowpack, are not represented?
We agree with the reviewer that there is limited interest in simulating the snowpack in forested areas in a remote sensing perspective. The difficulty of accurately modelling both the snowpack and radiative transfer under trees and snow makes for a particularly challenging problem. However, we have chosen to tackle the entire elevation range within our study area out of completeness, in order to assess how the subgridding framework is performing on the entire domain of the simulation."
This point remains quite unclear to me. As far as I understand, authors consider somehow "virtual open terrain" below treeline to cover a full elevation range. It should be more explicitly stated in the manuscript, together with motivations for doing so

We do not consider "virtual open terrain" for forested areas. We use the basic scheme for forest processes in SNOWPACK, and each forested cell according to the land-use classification is initiated with canopy information. However, we acknowledge the fact that forest-snow processes are a complex and the basic scheme in SNOWPACK is making a rough estimation of the reality. We added the following sentence in the Alpine3d paragraph of section 3.1 HRDPS subgridding and Alpine3D simulations:

Alpine3D uses a DEM and a land-use layer to properly initiate each SNOWPACK cell. Depending on the land-use category each cell falls into, canopy information is provided for forested cells in order to represent snow interception and forest snow processes.
(l. 166 in the revised manuscript)

"l. 271-272: "The wind erosion effect on the snowpack is also well represented, as dominant winds are blowing from the South / South-West. As a result, the south aspect profiles show more defragmented grains (dark green) on the surface". I am not sure I understand this cause-consequence. As far as I understood, wind-induced snow transport is represented by a precipitation multiplier. Consequently, associated effects of snowdrift on snow microstructure are not represented.Or am I missing something? Please clarify.

The word "erosion" here has not been used appropriately by the authors and is certainly the cause of the misunderstanding. Lines 271 - 274 in the original manuscript have been modified as such: The wind effect on the snowpack is also well represented in the simulations. Indeed, dominant winds are blowing from the South / South-West, and as a result southern slopes are affected by stronger winds (fig. 7). In the SNOWPACK model, grain type is a function of dendricity and sphericity, two parameters governed by the temperature gradient within the snowpack. As the surface temperature is altered by surface winds, precipitation particles (lime green) on the south aspects tend to metamorphose faster into decomposing and fragmented precipitation particles (dark green) than in the northern aspects, especially in the alpine."

The grain type could be affected by many other parameters. In the present state, this conclusion is not sufficiently backed..

The paragraph on the influence of wind on grain defragmentation has been removed from the manuscript.

**Comments about the track change manuscript**

**Abstract**

All comments were taken into account as suggested by the referee

**Introduction**

All comments were taken into account as suggested by the referee

**Study Area**

All comments were taken into account as suggested by the refereee

**The Numerical Weather Predictions downscaling processing chain design**

All comments were taken into account as suggested by the referee

**Results**

Figure 10 is not so easy to read: in particular, what are the x axis legends? Pixels "names"? Maybe the authors could think of a more "reader-friendly" figure conveying the same message

The figure x-axis corresponds to HRDPS cell names. We improved the figure's caption and description to give the reader a better understanding how what is shown in the figure. However, we kept it as boxplot because in our opinion this is the best way to show the increase of SWE spatial variability among each HRDPS cell throughout the season. The figure description has been modified as such:

Boxplot of the SWE modelled by the subgridding framework within each HRDPS cell in the early season and in the end of the season. The labels on the x-axis correspond to each HRDPS cell ID. The box spans the interquartile range (IQR), the line represents the median,

and the whiskers extend to the minimum and maximum value within 1.5 times the IQR. Outliers have been removed.

All other comments were taken into account as suggested by the referee

**Discussion**
All comments were taken into account as suggested by the referee
On unit choice for SWE: SWE was indeed written in cm (no mistake there). The unit was switched to mm to conform with a more commonly used unit.

**Conclusion**
All comments were taken into account as suggested by the referee